# The bimodally expressed microRNA miR-142 gates exit from pluripotency

Hanna L Sladitschek & Pierre A Neveu[*]

## Abstract

A stem cell's decision to self-renew or differentiate is thought to critically depend on signaling cues provided by its environment. It is unclear whether stem cells have the intrinsic capacity to control their responsiveness to environmental signals that can be fluctuating and noisy. Using a novel single-cell microRNA activity reporter, we show that miR-142 is bimodally expressed in embryonic stem cells, creating two states indistinguishable by pluripotency markers. A combination of modeling and quantitative experimental data revealed that mESCs switch stochastically between the two miR-142 states. We find that cells with high miR-142 expression are irresponsive to differentiation signals while cells with low miR-142 expression can respond to differentiation cues. We elucidate the molecular mechanism underpinning the bimodal regulation of miR-142 as a double-negative feedback loop between miR-142 and KRAS/ERK signaling and derive a quantitative description of this bistable system. miR-142 switches the activation status of key intracellular signaling pathways thereby locking cells in an undifferentiated state. This reveals a novel mechanism to maintain a stem cell reservoir buffered against fluctuating signaling environments.

**Keywords** microRNA; single-cell heterogeneity; stem cell differentiation; stochasticity

**Subject Categories** Development & Differentiation; Quantitative Biology & Dynamical Systems

**Mol Syst Biol. (2015) 11: 850**

## Introduction

Stem cells respond to internal and external cues by self-renewal or commitment to a differentiated fate (North *et al*, 2007; Jiang *et al*, 2009; Medema & Vermeulen, 2011; Kueh *et al*, 2013; Blanpain & Fuchs, 2014). Current models suggest that this balance is controlled *in vivo* by stem cell niches (Scadden, 2006; Voog & Jones, 2010; Simons & Clevers, 2011) and *in vitro* by an appropriate growth factor environment (Murry & Keller, 2008; Pera & Tam, 2010).

Mouse embryonic stem cells (mESCs) constitute a powerful system to study the molecular mechanism of fate decisions in controlled *in vitro* environment (Rué & Martinez Arias, 2015). mESCs are continuous cell lines derived from the inner cell mass of the blastocyst (Evans & Kaufman, 1981; Martin, 1981). These cells can be propagated indefinitely *in vitro* while maintaining their pluripotency, that is the capacity to give rise to derivatives of all three germ layers and germ cells both *in vitro* and *in vivo*.

microRNAs (miRNAs) are small non-coding RNAs that act as post-transcriptional regulators of gene expression (Bartel, 2009). A growing body of evidence suggests that miRNAs act as key players in stem cell homeostasis (Neumüller *et al*, 2008; Foronda *et al*, 2014) and cell fate decisions (Chen *et al*, 2004; Johnston *et al*, 2005; Li & Carthew, 2005; Wang *et al*, 2007; Yi *et al*, 2008; Schwamborn *et al*, 2009). Whereas the role of transcription factor heterogeneity in defining different pluripotent substates is well established (Chambers *et al*, 2007; Singh *et al*, 2007; Toyooka *et al*, 2008), it is largely unknown whether such dynamic heterogeneity exists at the level of miRNA expression.

To address this gap in our knowledge, we used a single-cell miRNA activity reporter to identify miR-142 that is bimodally expressed in mESCs under pluripotency-maintaining conditions. miR-142 expression levels stratify mESCs with indistinguishable expression of pluripotency markers into two distinct subpopulations: mESCs with low miR-142 levels are amenable to signal-induced differentiation, while cells with high miR-142 levels are irresponsive to differentiation cues. Using quantitative experiments and simulations, we show that mESCs switch stochastically between the high and low miR-142 states. Dissecting the molecular mechanism, we find that miR-142 represses the activation of KRAS/ERK signaling in a double-negative feedback loop that creates a bistable system. We propose that the self-generated miR-142 two-state system functions to maintain a stem cell reservoir that is protected from differentiation signals from the environment.

## Results

### miR-142 is a new marker of mESC heterogeneity under naïve pluripotency conditions

We reasoned that miRNAs that control different self-renewing mESC states should show heterogeneous expression under uniform

Cell Biology and Biophysics Unit, European Molecular Biology Laboratory, Heidelberg, Germany
*Corresponding author. Tel: +49 6221 387 8336; E-mail: neveu@embl.de

pluripotency-maintaining conditions. To identify such miRNAs, we devised a ratiometric fluorescence sensor that can visualize miRNA activity in single cells. The reporter consists of a bidirectional promoter driving the expression of a normalizer (H2B-mCherry) and a miRNA detector (H2B-Citrine), which contains in its 3′-UTR a target sequence of the miRNA of interest (Figs 1A and EV1A). Using this reporter system, we screened 33 conserved miRNAs associated with differentiation, pluripotency or cell proliferation (Fig EV1B and C) in mESC lines stably expressing specific reporters. As expected, we found the abundant miR-294 to be highly active, whereas the differentiation-associated miRNA let-7 showed little activity (Fig EV1D and E). Most miRNA reporters displayed a normally distributed cell-to-cell variation comparable to a non-targeted control. Strikingly, however, we found a strongly variegated activity of miR-142-3p that divided clonal mESC colonies into two sectors with very different miRNA activity (Fig 1B). Also at the population level, miR-142-3p activity was clearly bimodal distinguishing two mESC populations with either a high or a low miR-142-3p activity state (in the following referred to as "high" and "low" miR-142 states, Fig 1C and D). Furthermore, this bimodal regulation was present in chemically defined media conditions that support naïve pluripotency including "2i" but was absent in primed pluripotency (Fig EV2). Thus the bimodal regulation of miR-142 represents a novel kind of mESC heterogeneity in LIF-dependent pluripotency.

To validate that our reporter responded specifically to miR-142-3p, we generated $mir142^{-/-}$ mESC lines by deleting both alleles of $mir142$ using the CRISPR/Cas9 technology (Appendix Fig S1). As expected, the repression of the reporter was relieved in $mir142^{-/-}$ cells (Fig 1E). In addition, we assessed miRNA expression levels of FACS-purified "high" and "low" miR-142 state subpopulations in wild-type mESCs by deep sequencing. This analysis showed a 10-fold increase in the expression levels of miR-142-3p and miR-142-5p, the two mature forms of the miR-142 stem loop, in "high" miR-142 mESCs compared to "low" miR-142 mESCs (Fig 1F). Expression levels of all the other detected miRNAs were tightly correlated between the "high" and "low" miR-142 states (Fig 1F). Finally, we calibrated our single-cell miRNA activity reporter using expression data of miR-142-3p and mRNA reporter levels measured by deep sequencing. As expected, the signal of miR-142 activity reporter depends on miR-142-3p levels following Hill's equation with non-cooperative binding (Fig 1G). Our reporter thus specifically measures the activity of miR-142-3p. Low miR-142-3p levels correspond to large reporter ratios and high miR-142-3p levels yield small reporter ratios. The system therefore allows us to quantitate miR-142 expression changes in single living cells.

### The two miR-142 states are indistinguishable by pluripotency markers

Previous reports of mESC heterogeneity found a metastable coexistence of a pluripotent state and a state prone to differentiate and already expressing lower levels of the pluripotency markers Nanog and Rex1 (Chambers et al, 2007; Singh et al, 2007; Toyooka et al, 2008; Singer et al, 2014). To test if miR-142 heterogeneity is upstream of expression changes in pluripotency markers, we analyzed FACS-purified "high" and "low" miR-142 mESC populations. The two miR-142 states were both positive for alkaline phosphatase staining (Fig 2A). mRNA profiles measured by deep

sequencing of FACS-purified "high" and "low" miR-142 state mESCs clustered together, while the mRNA profiles of mESCs with low Nanog expression clustered apart (Fig 2B–D). In addition, the miRNA expression profile of mESCs with low Nanog expression was markedly different from the miRNA expression signature of "high" and "low" miR-142 states (Appendix Fig S2). Gene set enrichment analysis (Subramanian et al, 2005) using curated gene sets (canonical pathways, BioCarta and KEGG gene sets) did not reveal any significantly dysregulated gene set between the "high" and "low" miR-142 states. Comparing the expression of genes with a highly variable expression in mESCs (Klein et al, 2015), we found only 14 genes out of 1,891 with more than twofold expression changes between the two miR-142 sates (Appendix Fig S3). This number is similar to the one obtained when comparing biological replicates. In comparison, 302 genes display more than twofold changes in expression in low Nanog cells compared to high Nanog cells. Predicted targets of miR-142-3p had significantly lower expression in "high" miR-142 mESCs compared to "low" miR-142 mESCs ($P < 0.001$, determined by subsampling) and removing predicted miR-142-3p targets was sufficient to abrogate the clustering of "high" and "low" miR-142 expression profiles (Appendix Fig S4). Closer examination of pluripotency factor expression showed no significant difference in the mRNA or protein expression levels of Oct4 (or Pou5f1), Nanog, Rex1 (or Zfp42) and Sox2 between the "high" and "low" miR-142 states in FACS-purified subpopulations (Fig 2E and F). Furthermore, the "high" and "low" miR-142 states showed no difference in the known heterogeneity of NANOG (Figs 2G and EV3A) or REX1 (Fig EV3B) protein expression at the single-cell level. Moreover, all cells stained positive for the pluripotency markers OCT4 and SSEA-1 irrespective of their "high" or "low" miR-142 state identities (Fig EV3C and D). Additional pluripotency markers (Ng & Surani, 2011) showed no significant difference at the mRNA expression levels (Fig EV3E). In addition, neither "high" nor "low" miR-142 state cells shared molecular markers with epiblast stem cells (Fig EV3F), that reside in a state of primed pluripotency. Thus, the "high" and "low" miR-142 states are indistinguishable in their pluripotency marker expression and did not represent a primed pluripotent state.

Finally, we introduced the miR-142 activity reporter in a Rex1-dGFP knockin mESC line (Wray et al, 2011) in order to compare the bimodal regulation of miR-142 to the known heterogeneity in Rex1 expression. mESCs with high Rex1-dGFP levels revealed a bimodal regulation of miR-142 activity, that is mESCs with high Rex1-dGFP reside in either the "high" miR-142 state or the "low" miR-142 state (Fig 2H and I). Moreover, cells with low Rex1-dGFP expression had a unimodal reporter distribution with a reporter ratio comparable to $mir142^{-/-}$ mESCs, corresponding to an absence of miR-142 expression (Fig 2H and I). Thus, the "high" miR-142 state and the "low" miR-142 state are only found together in the high Rex1 mESC compartment. This finding places miR-142 bimodality upstream of the so far described heterogeneity in pluripotency transcription factor expression. Therefore, miR-142 bimodality represents a novel kind of heterogeneity in naïve mESCs.

### The two miR-142 states interconvert stochastically

To assess whether and how the two miR-142 states can interconvert into each other, we monitored the distribution of miR-142 activity

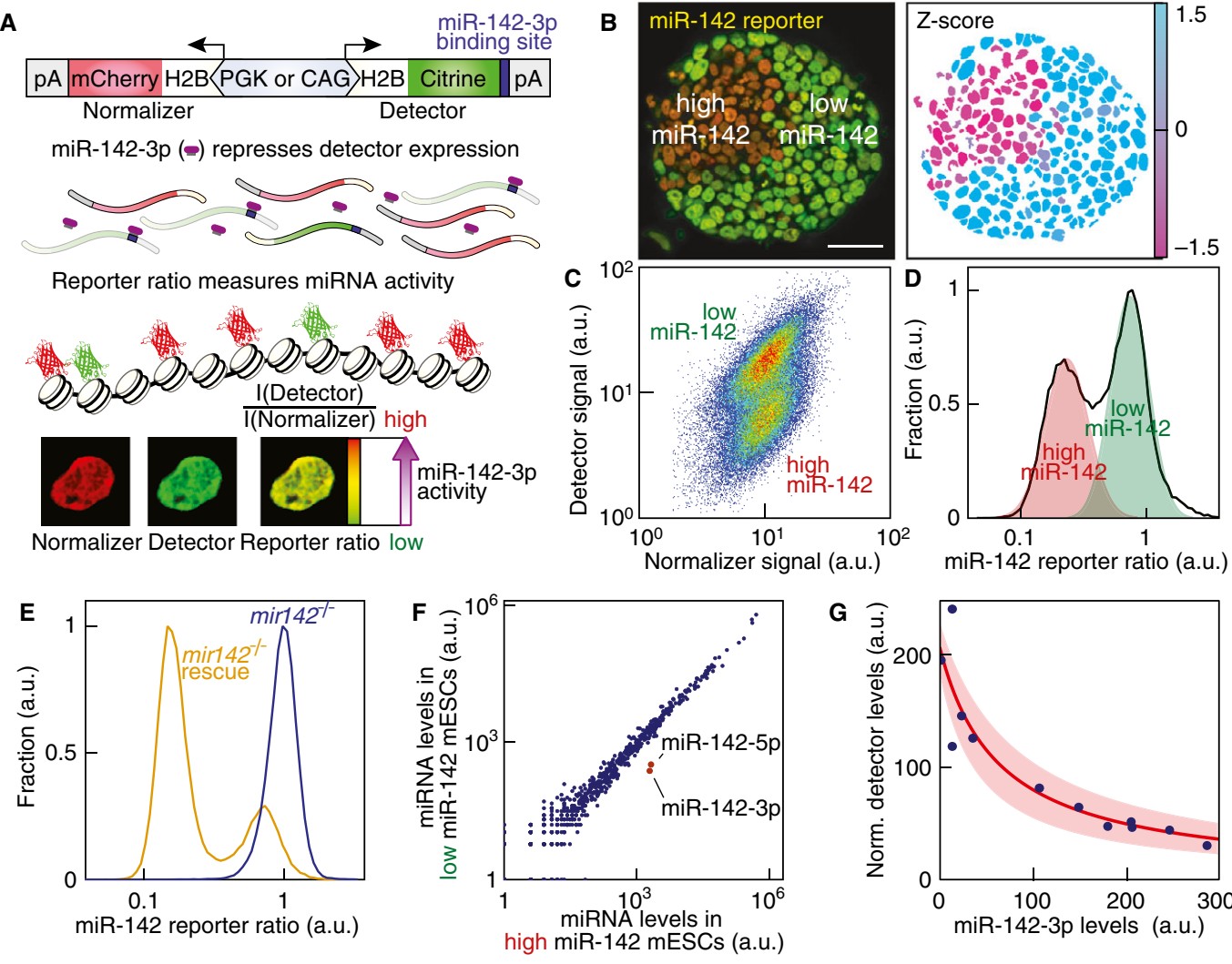

**Figure 1. The bimodal expression of miR-142 distinguishes two states in mESCs.**

A    Scheme of the experimental approach to monitor miRNA activity in single cells.

B    Confocal section of a single-cell-derived mESC colony stably expressing the miR-142-3p activity reporter and corresponding Z-score of the reporter ratio. As individual mESCs are not motile within a colony, sister lineages are spatially clustered. Scale bar: 50 µm.

C    Detector and normalizer expression in a clonal mESC population stably expressing the miR-142-3p reporter.

D    Distribution of the miR-142-3p reporter ratio in a clonal mESC culture. Two log-normal distributions ("high" miR-142 activity state: red shaded area; "low" miR-142 activity state: green shaded area) approximated well the experimental data (black line).

E    Distribution of the miR-142-3p reporter ratio in *mir142$^{-/-}$* mESCs (blue line) and *mir142$^{-/-}$* mESCs transgenic for the *mir142*-hosting lincRNA driven by its own promoter (*mir142$^{-/-}$* rescue, orange line; see Materials and Methods for details).

F    Deep sequencing analysis of miRNA expression levels in FACS-purified "high" and "low" miR-142 states. Levels of the two mature forms of miR-142, miR-142-3p and miR-142-5p are highlighted in red.

G    Adjustment of detector mRNA expression levels as a function of miR-142-3p levels (blue dots: experimental data measured by deep sequencing in FACS-purified populations) with Hill's equation with non-cooperative binding (red line; shaded area: fit confidence interval).

after FACS purification of "high" or "low" miR-142 subpopulations (Fig 3A). Indeed, either state could regenerate the other within 10 days of culture under pluripotency conditions (Fig 3A). State recovery was not due to any differential growth between the two miR-142 states because "high" and "low" miR-142 cells divided at the same rate every 12 h (Fig 3B-D). To determine the interconversion rates, we quantified the fraction of the population in the "high" and "low" miR-142 states. We then fitted the population data using first order reaction kinetics (see Materials and Methods for details,

Fig 3E and Appendix Fig S5A). Cells converted from "high" to "low" miR-142 states with a rate $k_1 = 0.072 \pm 0.01$ per cell division (on average one switching event every 14 divisions), while the backconversion was slightly slower occurring with a rate $k_{-1} = 0.048 \pm 0.006$ per cell division (on average one switching event every 21 divisions). Investigating the reporter ratio distribution in cultures derived from FACS-purified single cells showed that cultures recovered both states whether starting from "high" or "low" miR-142 founder cells (Appendix Fig S5B; $n > 160$). This

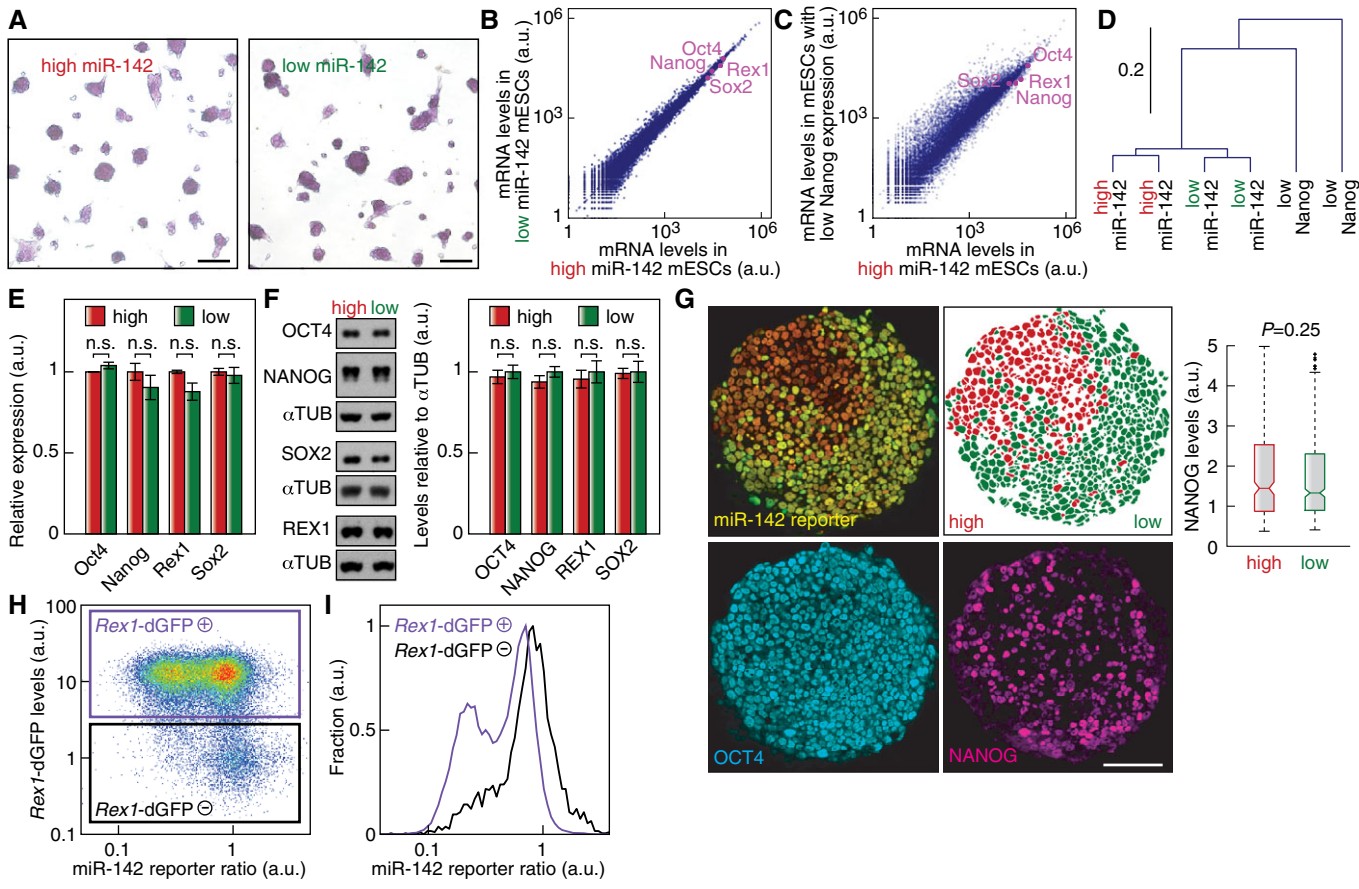

**Figure 2.** "High" and "low" miR-142 cells express pluripotency markers at equal levels.

A Alkaline phosphatase staining of FACS-purified "high" and "low" miR-142 state mESCs. Cells were cultured for 24 h after sorting and stained. Scale bar: 100 μm.

B, C Deep sequencing analysis of mRNA expression levels in FACS-purified "high" and "low" miR-142 mESCs (B) or mESCs with low Nanog expression (C).

D Average linkage hierarchical clustering of mRNA profiles of "high" and "low" miR-142 mESCs and of mESCs with low Nanog expression.

E mRNA expression levels of pluripotency markers in FACS-purified "high" and "low" miR-142 state mESCs (*n* = 2; n.s.: not significant, two-sided *t*-test). Data represented as mean ± SEM.

F Western blot analysis and quantification of pluripotency marker levels in FACS-purified "high" and "low" miR-142 state mESCs (*n* = 7; n.s.: not significant, two-sided *t*-test). Data represented as mean ± SEM.

G Immunostaining of OCT4 and NANOG in a clonal miR-142-3p reporter mESC colony. Quantification of NANOG levels in individual cells showed no significant difference in NANOG expression between the "high" and "low" miR-142 states (*P* = 0.25, Kolmogorov–Smirnov test). In the plot, the whiskers denote 1.5 times the interquartile range. Scale bar: 100 μm.

H miR-142 activity reporter ratio in a *Rex1*-dGFP knockin mESC line.

I Distribution of miR-142 reporter ratio in cells positive for *Rex1*-dGFP expression (*Rex1*-dGFP⁺, purple line) and negative for *Rex1*-dGFP expression (*Rex1*-dGFP⁻, black line). Gates identifying the populations are displayed in (H).

demonstrated that all clonogenic cells can switch between states. Single-cell live imaging of miR-142 activity revealed that switching occurred rapidly within less than a cell cycle and that after division sister lineages were not always correlated in their switching behavior (Fig 3F and G, and Movie EV1) suggesting stochastic switching events.

If switching were indeed stochastic, the variegated distribution of "high"/"low" miR-142 cells in a colony grown from a single cell will depend on the time when the first state switching occurred, since the states are on average stable for several cell cycles (Fig 4A). Using a stochastic switching model, we could simulate the expected fraction of cells that switched state in colonies derived from pure "high" or "low" miR-142 state single founders (Fig 4B). To test this prediction, we measured the fraction of switched cells in colonies grown from single FACS-purified "high"

or "low" miR-142 cells. The stochastic switching model approximated well data from founder cells FACS-purified in the "low" miR-142 state but could not fit with the same parameters the state composition obtained in cultures derived from founder cells FACS-purified in the "high" miR-142 state (Fig 4B). We thus introduced a refined model in which cells stochastically switch between the two miR-142 states but "high" and "low" miR-142 cells can have different survival rates under clonogenic conditions while having the same proliferation rate (Fig 4C). The experimental data were recapitulated by simulations using this refined model including a survival bias for "low" miR-142 cells under clonogenic conditions (see Materials and Methods for details, Fig 4D). Interestingly, we experimentally confirmed this survival bias predicted by the model. Indeed, clonogenicity of FACS-purified "low" miR-142 cells (19.8 ± 6.6%) was higher than for "high" miR-142 cells

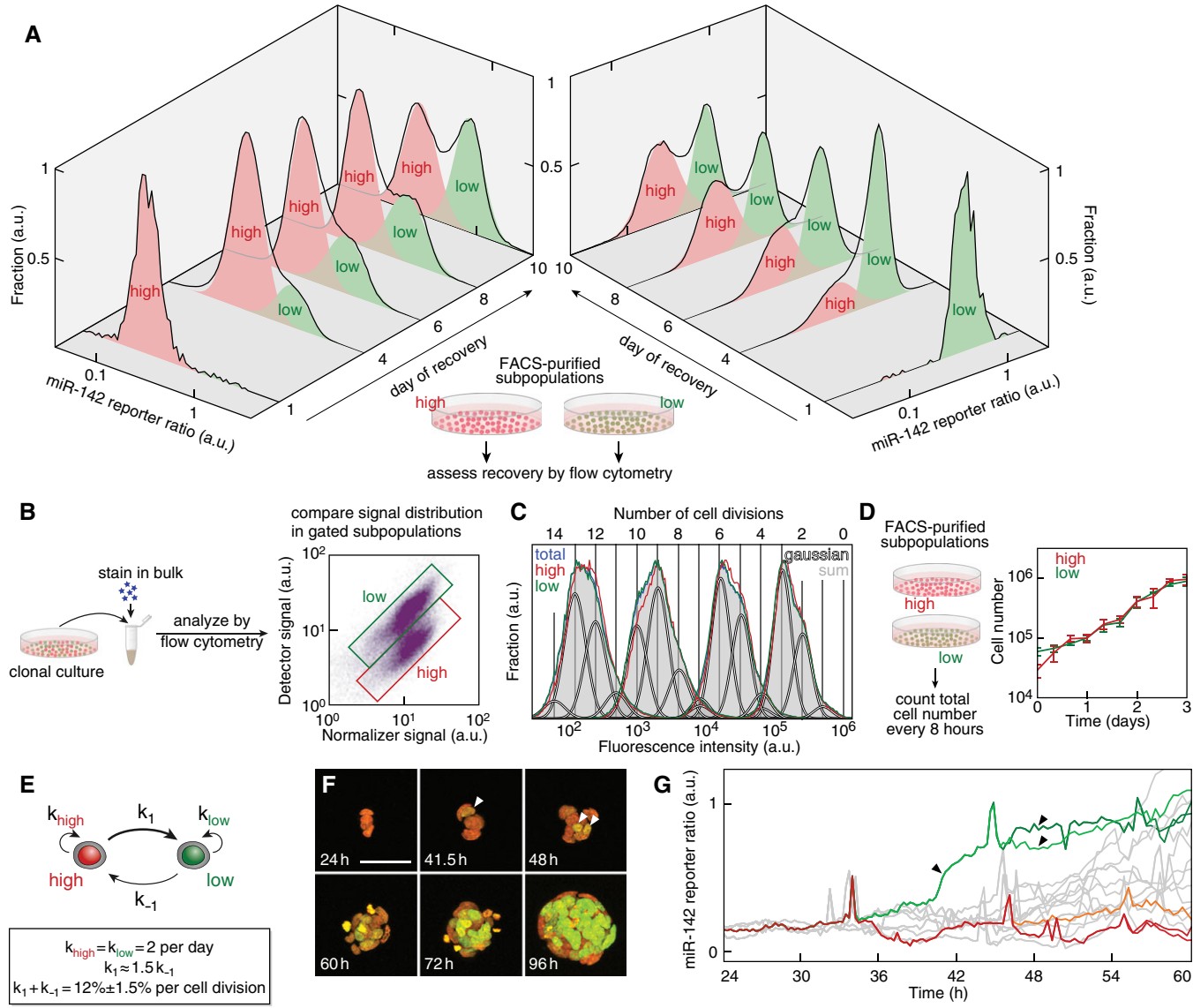

**Figure 3. Interconversion between the two miR-142 states.**

A  "High" and "low" miR-142 subpopulations were FACS-purified and the temporal evolution of the miR-142 reporter ratio was measured. Shaded areas: quantification of cells in "high" or "low" miR-142 states.

B  Scheme of the experimental design to compare proliferation rates of mESCs in the "high" and "low" miR-142 state using a dye dilution by cell division strategy.

C  Time-lapse analysis of the fluorescence intensity of cells stained on day 0 with a commercial dye labeling free amines. The mESC culture was analyzed by flow cytometry each day. The distribution of the dye retention in all live cells in the culture (blue line) could be well approximated by the sum (shaded gray area) of gaussian distributions (white lines outlined in black) representing distinct cell division cycles. The distribution of dye retention in mESCs in the "high" or "low" miR-142 state is outlined by a red or green line.

D  Population growth of mESCs starting from FACS-purified "high" (red line) and "low" (green line) miR-142 subpopulations (error bars represent SEM, $n = 6$).

E  Reaction kinetics model of the interconversion between "high" and "low" miR-142 cells. $k_{high}$ and $k_{low}$ are the proliferation rates of "high" and "low" miR-142 cells, $k_1$ the interconversion rate from "high" to "low" miR-142 state and $k_{-1}$ the interconversion rate from "low" to "high" miR-142 state.

F  Live imaging of switching events during the growth of a single-cell-derived mESC colony. Maximal projections of confocal stacks are shown at the indicated time points (h: hour). White arrowheads denote the cell with the first switching event at 41.5 h and the two resulting daughters at 48 h. Scale bar: 50 μm. See also Movie EV1.

G  Single-cell tracks of reporter ratio in two sister lineages (green: sister lineage with activity switching, red and orange: sister lineage without activity switching, gray: all other lineages; spikes in reporter signal are artifacts due to signal saturation at mitotic divisions, black arrowheads correspond to the cells marked by white arrowheads in F).

(5.8 ± 3.1%) (Fig 4E). This gave a survival bias for "low" miR-142 cells of 6.9 (90% confidence interval: 1.8–17.8), in excellent agreement with the 8-fold survival bias predicted by the simulations. Using a genetic loss-of-function approach, we could show that the loss of *mir142* expression indeed improved clonogenicity without affecting the proliferation rate (Fig 4F). In summary, we could demonstrate experimentally and theoretically that individual mESCs fluctuate stochastically between the two miR-142 states at

a relatively low rate with a state switching event occurring on average every 8 cell divisions.

## Constitutive miR-142 expression locks cells in an undifferentiated state

A hallmark of embryonic stem cells is the ability to generate distinct differentiated cell types. To assess whether *mir142* expression affects differentiation capacity, we compared *mir142* gain- or loss-of-function mESCs regarding their capabilities to differentiate toward fates of the three germ layers, that is neuroectoderm, mesoderm, and endoderm fate. Upon differentiation, $mir142^{-/-}$ cells stained positive for the neuronal marker Tuj1 (or βIII-tubulin), the muscle marker Desmin or the endoderm marker Foxa2 and were negative for the pluripotency marker Oct4 (Fig 5A–C and

Appendix Fig S6). By contrast, *mir142* gain-of-function cells retained Oct4 expression and showed no differentiation marker expression (Fig 5D–F and Appendix Fig S6). In order to understand genome-wide this striking difference in response to differentiation cues, we profiled the transcriptomes of wild-type mESCs, $mir142^{-/-}$ and *mir142*-expressing mESCs during a 6 day endoderm differentiation time course. Strikingly, cells constitutively expressing *mir142* always clustered with undifferentiated wild-type and $mir142^{-/-}$ cells at day 1 or 2 (Fig 5G), while differentiating wild-type mESCs and $mir142^{-/-}$ cells from day 3–6 cluster separately. Using principal component analysis allowed us to visualize the trajectory of expression profiles during 6 days of differentiation (Fig 5H). This showed that unlike wild-type and $mir142^{-/-}$ cells, cells constitutively expressing *mir142* were essentially locked in an undifferentiated expression state (Fig 5H and Appendix Fig S7A and B) and

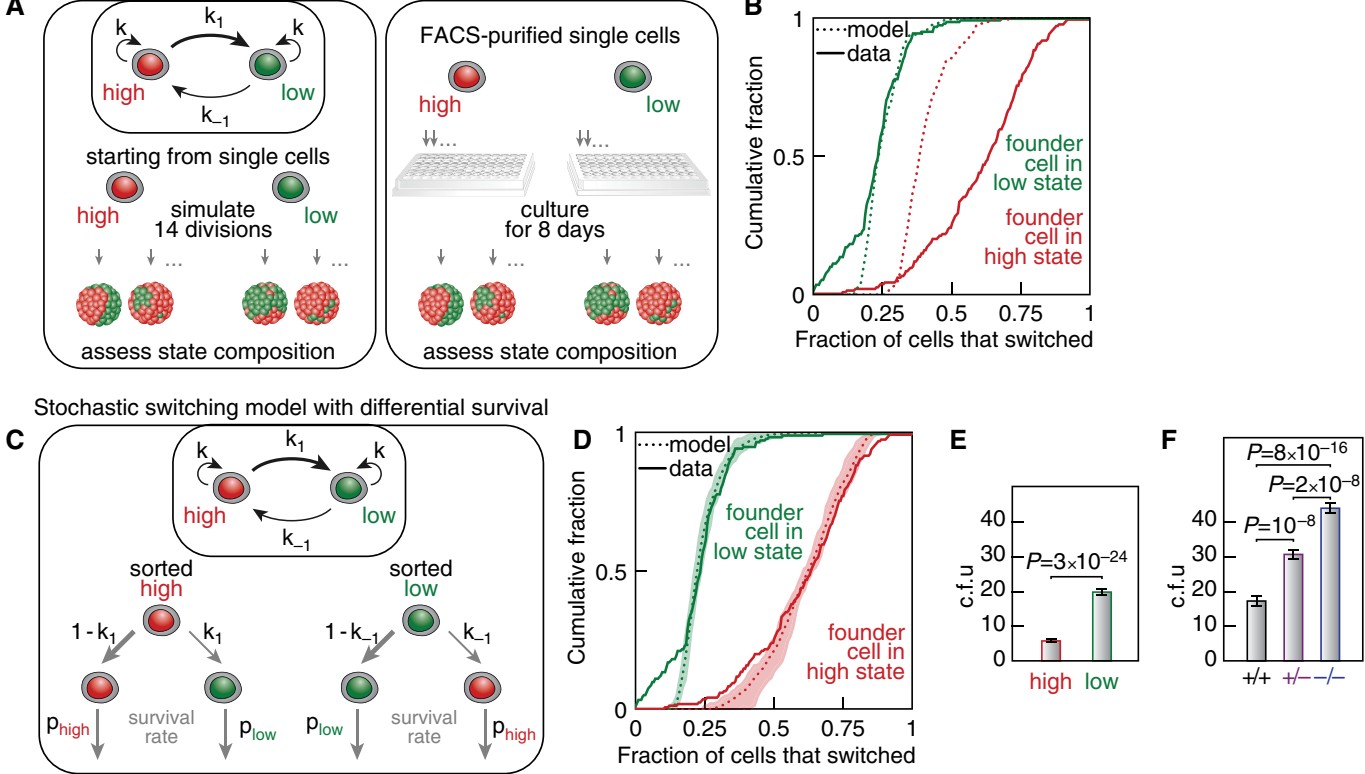

**Figure 4. mESCs switch stochastically between the two miR-142 states.**

A   Stochastic switching model (left panel): Cells can switch state each cell division with probability $k_1$ or $k_{-1}$. Experimental scheme (right panel): Clonal cultures were derived from single FACS-purified "high" and "low" miR-142 mESCs. Occurrence of switching events was measured by assessing the miR-142 reporter ratio distribution in individual cultures.

B   Simulation of state distribution after colony growth following the model shown in (A) with a founder cell in "low" miR-142 (dashed green line) or "high" miR-142 (dashed red line) state (170 colonies, 14 divisions, $k_1 + k_{-1} = 0.08$ per cell division, $k_1 = 1.5\ k_{-1}$). Solid red and green lines: experimentally measured state distribution in cultures derived from FACS-purified founder cells in "high" and "low" miR-142 states ($n = 169$ and $n = 171$).

C   Stochastic switching model with differential survival. "High" and "low" miR-142 mESCs can have different survival rate under clonogenic conditions.

D   Simulation of state distribution after colony growth following the stochastic switching model with differential survival shown in (C) with a founder cell in "low" miR-142 (dotted green line) or "high" miR-142 (dotted red line) state (170 colonies, 14 divisions, $k_1 + k_{-1} = 0.08$ per cell division, $k_1 = 1.5\ k_{-1}$, $p_{low}/p_{high} = 8$). Shaded area: 95% confidence interval. Solid red and green lines: experimental data for FACS-purified founder cells in "high" and "low" miR-142 states, same data as shown in (B).

E   Clonogenicity of single FACS-purified founder cells in "high" or "low" miR-142 state (c.f.u.: colony forming units; $n = 50$; $P = 3 \times 10^{-24}$, two-sided $t$-test; error bars represent SEM). Single cells were FACS-purified in 96-well plates ($n$ represents the number of 96-well plates that were analyzed).

F   Clonogenicity of single $mir142^{+/+}$ ($n = 19$), $mir142^{+/-}$ ($n = 20$) or $mir142^{-/-}$ ($n = 20$) mESCs (c.f.u.: colony forming units; $P = 10^{-8}$, $P = 2 \times 10^{-8}$ and $P = 8 \times 10^{-16}$, two-sided $t$-test; error bars represent SEM). Single cells were FACS-purified in 96-well plates ($n$ represents the number of 96-well plates that were analyzed).

                    

consistently failed to up-regulate established endoderm markers (Fig 5I). Even at the end of the 6 day differentiation procedure, cells with constitutive *mir142* expression proliferated normally under pluripotency conditions, exhibited the characteristic 3-dimensional morphology of undifferentiated mESC colonies and were alkaline phosphatase-positive (Fig 5J). In addition, genetic deletion of *mir142* led to significantly larger changes in gene expression compared to wild-type cells as measured by projection on PC1 and PC2 (Appendix Fig S7A and B). Indeed, *mir142*$^{-/-}$ cells exhibited significantly higher levels of differentiation markers and a lower expression of pluripotency markers compared to wild-type cells at day 6 of differentiation (Appendix Fig S7C). Together, our data demonstrate that *mir142* expression locks mESCs in an undifferentiated state even if exposed to strong differentiation cues for several days.

### The "high" *mir142* subpopulation is delayed in differentiation

To test whether the naturally generated "high" miR-142 state also locks cells in an undifferentiated state, we differentiated wild-type mESCs expressing the miR-142 reporter toward neuroectoderm, mesoderm and endoderm fate. Upon differentiation toward neuroectoderm, mesoderm and endoderm fate, cells with "low" miR-142 activity stained positive for the neuronal marker Tuj1 (or βIII-tubulin), the muscle marker Desmin or the endoderm marker Foxa2, respectively (Fig 6A–C). In contrast, cells exhibiting "high" miR-142 activity stained positive for the pluripotency marker Oct4 independently of the differentiation regime (Fig 6A–C and Appendix Fig S8). We next aimed to characterize the effect of the endogenous bimodal miR-142 expression during the differentiation of a wild-type mESC population in more details. We first monitored the changes of miR-142 activity in FACS-purified "high" or "low" miR-142 cell populations undergoing differentiation to neuroectoderm, mesoderm and endoderm. "High" miR-142 cells gradually lost miR-142 activity (and *mir142* expression) over the first 4 days of differentiation irrespective of the differentiation regime, becoming in majority converted into a "low" miR-142 state by day 7 (Fig 6D and Appendix Fig S9). In contrast, "low" miR-142 cells did not change their miR-142 activity state under any differentiation cue (Fig 6E). Inspection of the differentiating cultures revealed that "high" miR-142 cells kept their 3-dimensional morphology up to 4 days, whereas "low" miR-142 cells readily adapted a differentiated monolayer morphology within 3 days (Fig 6F).

We next aimed to gain a genomewide view of the differences in gene expression in differentiating mESCs depending on their *mir142* levels. To do so, we subjected a bimodal mESC population grown under pluripotency conditions to differentiation cues for 3 days, FACS-purified cell populations with either high or low *mir142* expression and assessed gene expression by transcriptome profiling. Cells that exhibited high *mir142* levels after 3 days of differentiation clustered with undifferentiated mESCs, whereas cells with no *mir142* expression clustered with differentiated cells (Fig 6G). Projection onto the differentiation gene expression trajectory of wild-type mESCs confirmed that *mir142*-expressing cells remained at the beginning of the differentiation trajectory, while the profiles of *mir142*-negative cells had progressed to the end of the trajectory similar to differentiated cells (Fig 6H). Probing specific genes showed that *mir142*-expressing cells failed to down-regulate pluripotency genes in response to instructive differentiation cues (Fig 6I). To conclude, the naturally generated "high" miR-142 state locked cells in an undifferentiated state and the bimodal regulation of *mir142* establishes a dichotomy between a subpopulation amenable to differentiation (the "low" miR-142 cells) and a pool of cells delayed in differentiation (the "high" miR-142 cells).

### miR-142 states differ in AKT and ERK activation

Next, we sought a mechanistic understanding of the bimodal expression of miR-142 under LIF-dependent pluripotency conditions. Deep sequencing and a single-cell *mir142* transcriptional reporter demonstrated that transcriptional regulation accounted for the bimodal regulation of miR-142 activity (Fig EV4A and B). LIF is known to activate three distinct signaling pathways through its coreceptor gp130 (Fig 7A): the JAK/STAT3 (Niwa *et al*, 1998) and PI3K/AKT (Paling *et al*, 2004) pathways and the MEK/ERK signaling cascade (Burdon *et al*, 1999). Inhibiting ERK activity increased the expression of the *mir142* transcriptional reporter in a dose-dependent manner, while inhibition of STAT3 or activation of AKT had no effect (Figs 7B and EV4C). This indicated that ERK activity normally represses *mir142* expression, predicting that cells with "high" miR-142 activity have reduced ERK activity. To test this, we measured phosphorylated ERK kinase in FACS-purified "high" and "low" miR-142 state subpopulations. "High" miR-142 cells indeed showed reduced level of active ERK kinase compared to "low" miR-142 mESCs, while total ERK levels were unaffected ($P = 7 \times 10^{-7}$ for p-ERK, $n = 5$, two-sided *t*-test, Fig 7C). In addition phosphorylated AKT levels were reduced in "high" miR-142 cells compared to "low" miR-142 mESCs, while total AKT levels were unaffected as was phosphorylated and total STAT3 ($P = 8 \times 10^{-9}$ and $P = 0.78$ for p-AKT and p-STAT3, $n = 5$, two-sided *t*-test, Fig 7C). Thus, the miR-142 states correspond to two subpopulations of mESCs that differed in the activation status of ERK and AKT signaling.

### miR-142 balances AKT and ERK activation

The surprising finding that the LIF-stimulated signaling pathways ERK and AKT have different activity states in cells with "high" or "low" miR-142 activity despite being exposed to the same amount of LIF, prompted us to ask whether miR-142 itself tunes the sensitivity to LIF by repressing upstream signaling components of ERK and AKT. Consistent with bioinformatic prediction (Grimson *et al*, 2007), we found gp130 and Kras to be direct targets of miR-142-5p and miR-142-3p in mESCs (Appendix Fig S10). In addition, gp130 and Ras protein levels were significantly down-regulated in "high" compared to "low" miR-142 mESCs (Fig 7D). Furthermore, *mir142*$^{-/-}$ mESCs had elevated p-ERK and p-AKT levels as well as increased gp130 and RAS protein levels compared to *mir142*$^{+/+}$ mESCs ($P = 2 \times 10^{-7}$, $P = 10^{-4}$, $P = 0.038$ and $P = 0.003$ respectively, $n = 4$, two-sided *t*-test, Fig 7E). These instrumental effects of *mir142* on ERK and AKT activation and gp130 and RAS protein expression were confirmed using a gain-of-function approach (Fig EV5). Altogether our results support a model where miR-142 represses gp130 and Ras expression, thereby reducing the transduction of LIF into ERK activity. This, in turn, relieves the ERK-dependent repression of *mir142* expression, therefore forming a double-negative feedback loop (Fig 7F).

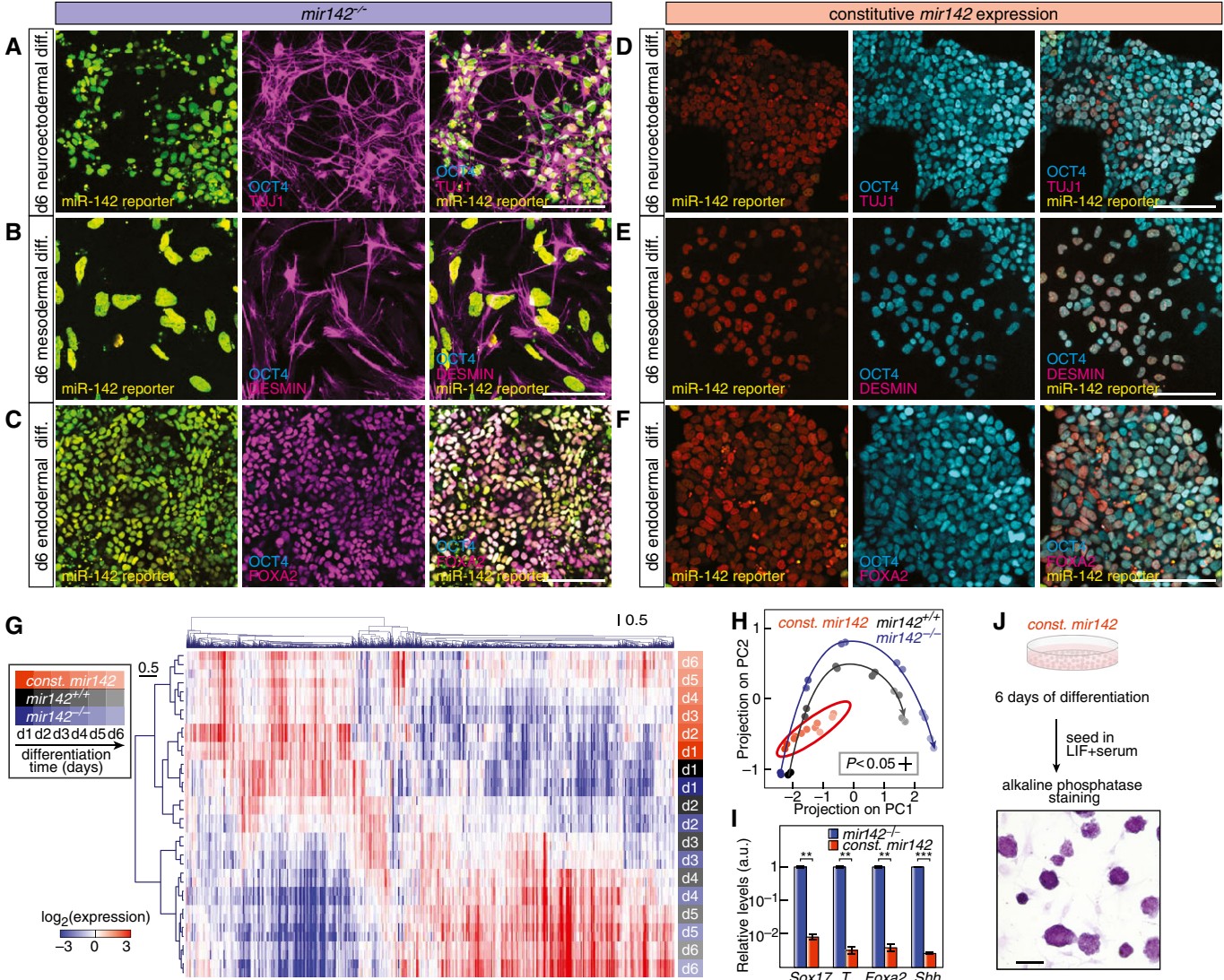

**Figure 5.  *mir142* expression locks mESCs in an undifferentiated state.**

A   miR-142 reporter signal (left panel) and immunostaining of TUJ1 and OCT4 (middle panel) in *mir142⁻/⁻* mESCs differentiated for 6 days to neuroectoderm. Scale bar: 100 μm.

B   miR-142 reporter signal (left panel) and immunostaining of DESMIN and OCT4 (middle panel) in *mir142⁻/⁻* mESCs differentiated for 6 days to mesoderm. Scale bar: 100 μm.

C   miR-142 reporter signal (left panel) and immunostaining of FOXA2 and OCT4 (middle panel) in *mir142⁻/⁻* mESCs differentiated for 6 days to endoderm. Scale bar: 100 μm.

D   miR-142 reporter signal (left panel) and immunostaining of TUJ1 and OCT4 (middle panel) in mESCs with constitutive *mir142*-expression differentiated for 6 days to neuroectoderm. Scale bar: 100 μm.

E   miR-142 reporter signal (left panel) and immunostaining of DESMIN and OCT4 (middle panel) in mESCs with constitutive *mir142*-expression differentiated for 6 days to mesoderm. Scale bar: 100 μm.

F   miR-142 reporter signal (left panel) and immunostaining of FOXA2 and OCT4 (middle panel) in mESCs with constitutive *mir142*-expression differentiated for 6 days to endoderm. Scale bar: 100 μm.

G   Hierarchical clustering of mRNA expression in *mir142⁺/⁺*, *mir142⁻/⁻* mESCs or mESCs with constitutive *mir142*-expression (const. *mir142*) during differentiation to endoderm progenitors (*mir142⁺/⁺*: black; *mir142⁻/⁻*: blue; const. *mir142*: orange; shading denotes the day of differentiation according to the boxed legend).

H   Principal component analysis of genomewide mRNA expression during differentiation of *mir142⁺/⁺* (black dots), *mir142⁻/⁻* (blue dots) mESCs and mESCs with constitutive *mir142*-expression (const. *mir142*, orange dots). PC1 accounting for 60.6% of the variation was characterized by contributions of pluripotency and endoderm differentiation-associated genes, while PC2 (13.4% of the variation) was contributed by genes subjected to transient up- or down-regulation during the differentiation process. Shading denotes the day of differentiation according to the boxed legend in (G). Black and blue arrows depict the differentiation trajectory of *mir142⁺/⁺* and *mir142⁻/⁻* cells, respectively.

I    Endoderm marker expression in *mir142⁻/⁻* (blue) cells and cells with constitutive *mir142*-expression (orange) differentiated for 6 days (*n* = 2, **P < 0.01, ***P < 0.001, two-sided t-test). Data represented as mean ± SEM.

J    mESCs with constitutive *mir142*-expression (const. *mir142*) were differentiated for 6 days, replated in pluripotency-maintaining conditions and stained for alkaline phosphatase. Scale bar: 100 μm.

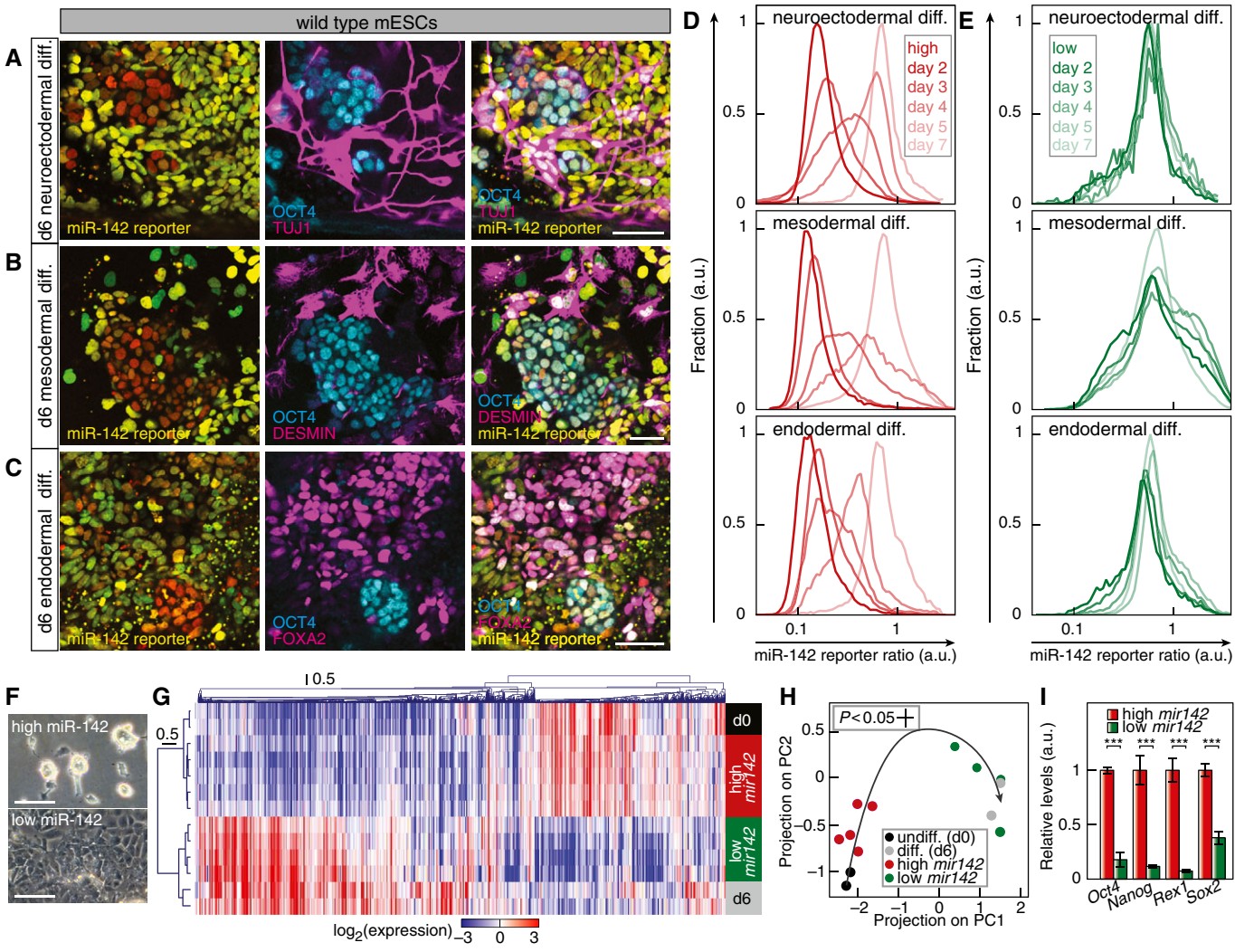

**Figure 6.** **The *mir142*-expressing subpopulation is delayed in differentiation.**

A   miR-142 reporter signal (left panel) and immunostaining of TUJ1 and OCT4 (middle panel) in wild-type mESCs differentiated for 6 days to neuroectoderm. Scale bar: 50 μm.

B   miR-142 reporter signal (left panel) and immunostaining of DESMIN and OCT4 (middle panel) in wild-type mESCs differentiated for 6 days to mesoderm. Scale bar: 50 μm.

C   miR-142 reporter signal (left panel) and immunostaining of FOXA2 and OCT4 (middle panel) in wild type mESCs differentiated for 6 days to endoderm. Scale bar: 50 μm.

D, E   Distribution of miR-142 activity reporter expression in FACS-purified "high" miR-142 cells (D) and FACS-purified "low" miR-142 cells (E) differentiated to neuroectoderm, mesoderm and endoderm. Shading denotes the time course of differentiation according to the boxed legend in the top panel.

F   Morphology of "high" and "low" miR-142 cells exposed for 3 days to endoderm differentiation cues. Scale bar: 100 μm.

G   Hierarchical clustering of mRNA expression of undifferentiated wild-type mESCs (d0, black), wild-type mESCs differentiated for 6 days (d6, gray) or subpopulations at day 3 of differentiation sorted according to their *mir142* levels (red: high *mir142*; green: low *mir142*).

H   Projection on the first two principal components PC1 and PC2 of mRNA expression profiles of high *mir142* (red dots) and low *mir142* (green dots) at day 3 of differentiation (black dots: undifferentiated mESCs at day 0; gray dots: differentiated cells at day 6). The black arrow represents the differentiation trajectory of wild-type cells using data from Fig 5H.

I   Pluripotency marker expression in high *mir142* and low *mir142* cells at day 3 of differentiation ($n = 5$ and $n = 4$, respectively, ***$P < 0.001$, two-sided *t*-test). Data represented as mean $\pm$ SD.

## The miR-142–ERK double-negative feedback loop creates a bistable system

If the double-negative miR-142–ERK feedback loop can establish a bistable system, it would provide the molecular explanation for the two miR-142 activity states that define differentiation-competent

and pluripotency-locked cells. We therefore used a simple kinetic model of non-linear feedback loops, which predicts a bistable region in the parameter space of miR-142 turnover rate ($\alpha$) and ERK activity turnover rate ($\gamma$) (Fig 7G and H). It is possible to find two steady states for $\beta > 2$ ($\beta$ corresponds to the ratio of total ERK to the affinity of its repressive interaction) with the size of the bistability region

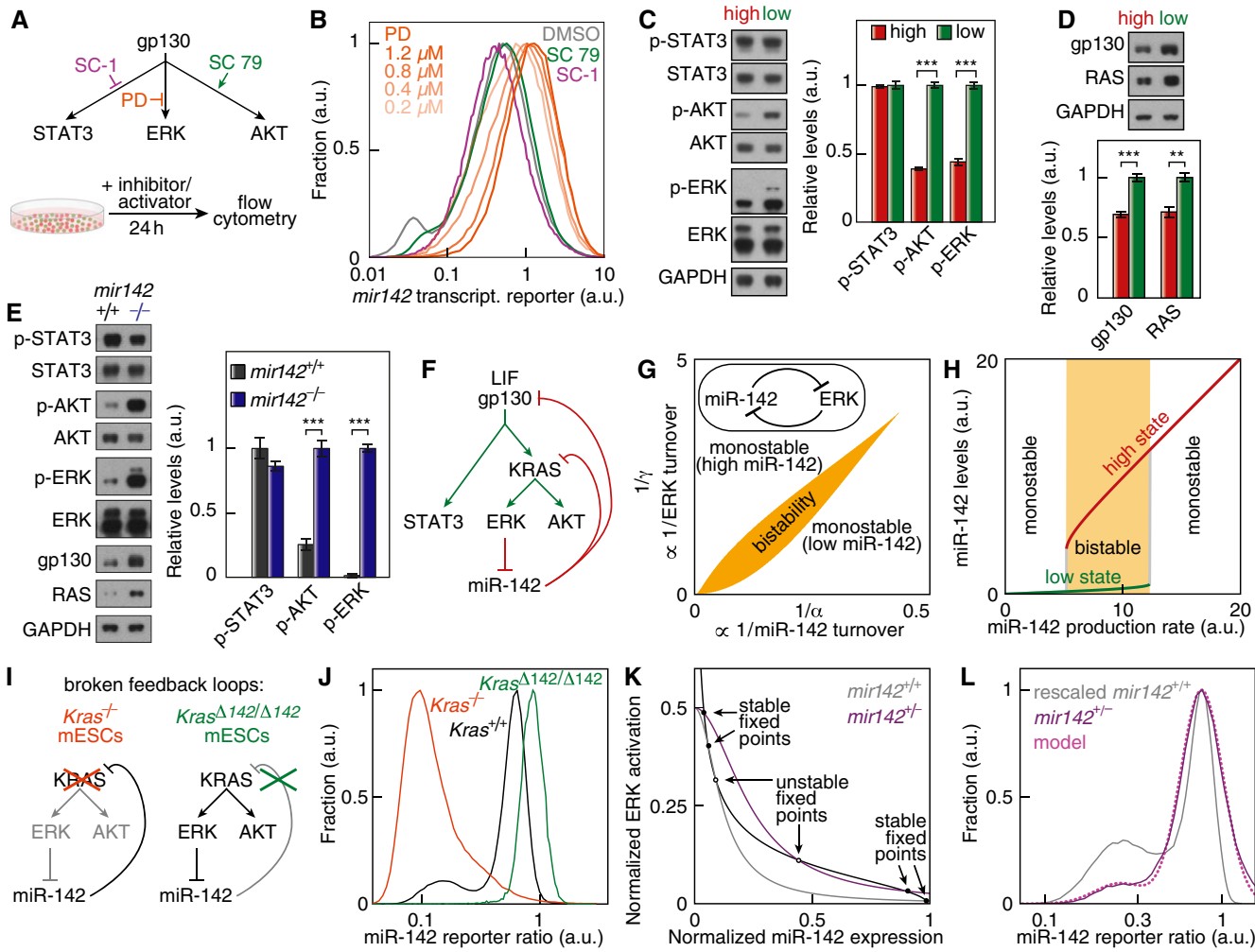

**Figure 7. Ras/ERK signaling and miR-142 form a double-negative feedback loop and produce a bistable system.**

A   Pharmacological interrogation of gp130-stimulated pathways. SC 79 activates AKT, SC-1 inhibits STAT3 activation and PD0325901 (PD) inhibits ERK activation.

B   Distribution *mir142* transcriptional reporter expression under pharmacological interrogation of gp130-stimulated pathways (orange lines shaded according to the concentrations shown in the panel: PD; green line: SC 79; magenta line: SC-1; gray line: DMSO control).

C   Activation status of ERK, AKT, STAT3 and quantification in FACS-purified "high" and "low" miR-142 mESCs (n = 5, ***P < 0.001, two-sided *t*-test). Data represented as mean ± SEM.

D   Protein levels of gp130 and RAS and quantification in FACS-purified "high" and "low" miR-142 mESCs (n = 5; **P < 0.01, ***P < 0.001, two-sided *t*-test). Data represented as mean ± SEM.

E   Activation status of ERK, AKT, STAT3 and protein levels of gp130 and RAS in *mir142*[+/+] and *mir142*[−/−] mESCs and quantification (n = 4; ***P < 0.001, two-sided t-test; error bars represent SEM).

F   miR-142 and Ras/ERK signaling form a double-negative feedback loop.

G   Theoretical phase diagram of the miR-142–ERK double-negative feedback loop. α represents a rescaled miR-142 turnover and γ an ERK activation turnover. Wild-type mESCs sit in the bistability region (orange).

H   Simulated miR-142 levels depending on miR-142 production rate. At low production rates, there exists a single low miR-142 state. For intermediate production rates, two stable high and low miR-142 states coexist. Finally a single high miR-142 state is found at high miR-142 production rates.

I   Design of systems with broken feedback loops through the deletion of *Kras* (*Kras*[−/−] mESCs) or the deletion of miR-142 binding sites in the 3′-UTR of *Kras* (*Kras*[Δ142/Δ142] mESCs).

J   Distribution of the miR-142 reporter ratio in *Kras*[−/−] mESCs (orange line), *Kras*[Δ142/Δ142] mESCs (green line) and wild-type mESCs (*Kras*[+/+], black line).

K   Nullclines of the miR-142 and ERK signaling double-negative feedback loop (black line: nullcline corresponding to *mir142*[+/+] cells; purple line: nullcline corresponding to *mir142*[+/−] cells).

L   Prediction of *mir142* expression levels in *mir142*[+/−] mESCs. Using the calibration of the reporter ratio response shown in Fig 1G, we could rescale the *mir142*[+/+] distribution (gray line) to derive predicted values of miR-142 concentration in the two stable miR-142 states. The reporter ratio distribution in *mir142*[+/−] mESCs (purple line) could be well approximated by the model (pink dotted line, fitting parameters: state occupancies).

in the (α, γ) parameter space increasing with increasing β (Appendix Fig S11). The aforementioned condition corresponds to total ERK levels being in excess compared to the levels necessary to

exert its repressive interaction. Bistability occurs mostly for large values for α, corresponding to small degradation rates compared to miR-142 production rates. Indeed, miRNAs are stable in a wide

variety of cells including mESCs (Krol *et al*, 2010), miRNA decay being mostly contributed by cell division (Gantier *et al*, 2011). Finally, bistability occurs mostly for large values for γ (of the order of one or greater). This corresponds to phosphorylation rates that are large compared to dephosphorylation rates, agreeing with experimental evidence (Fujioka *et al*, 2006). In conclusion, the parameter values necessary to observe bistability are in accordance with biological constraints.

As a first test of this bistable model, we broke the feedback loop by knocking out *Kras* in mESCs expressing the miR-142 activity reporter (Fig 7I and Appendix Fig S12). The model predicts that removing Kras would lead to uniformly high miR-142 expression, which we observed experimentally in $Kras^{-/-}$ mESCs (Fig 7J). As a further test of the implication of KRAS in the feedback loop, we deleted the miR-142 binding sites in the 3′-UTR of *Kras* (Fig 7I). The model predicts that removing those binding sites will lead to a uniformly low miR-142 expression, which we observed experimentally in $Kras^{\Delta142/\Delta142}$ mESCs (Fig 7J). This indicates that KRAS is a key player in establishing the bimodal expression of miR-142 in mESCs. As a second test whether mESCs follow the prediction of this bistability model, we used our $mir142^{+/-}$ cells, which possess a single *mir142* allele and thus half the gene dosage compared to wild-type cells and measured miR-142 regulation in LIF-dependent pluripotency conditions. The stable points of the miR-142–ERK dynamical system can be visualized by drawing the nullclines which are the curves along which the time derivatives of miR-142 or active ERK are equal to zero. The nullclines for both $mir142^{+/+}$ and $mir142^{+/-}$ cells indeed exhibited two stable fixed points for the same parameters (Fig 7K): one at high miR-142 levels and low ERK activation status and one at low miR-142 levels and high ERK activation status. Thus the bistable miR-142-ERK feedback system allowed *mir142* to be bimodally regulated in $mir142^{+/-}$ cells, which we observed experimentally (Appendix Fig S13). Moreover, the prediction of the two stable *mir142* expression levels in $mir142^{+/-}$ cells was in excellent agreement with the experimental data (Fig 7L). In conclusion, the double-negative feedback loop between miR-142 and Kras/ERK signaling creates a bistable system with a high miR-142 state with low ERK activation and a low miR-142 state with high ERK activation.

## Discussion

Our results support a model in which a double-negative feedback loop between Kras/ERK signaling and miR-142 produces a bistable system trapping mESCs in either a "low" miR-142 state that has a high level of ERK/AKT activity and is competent to differentiate or on the other hand a "high" miR-142 state that has a low level of ERK/AKT activity and is blocked from differentiation. High miR-142 levels hereby act as a red traffic light that prevents cells from transducing differentiation signals into gene expression programs driving exit from pluripotency and initiation of differentiation (Fig 8A and B). Low miR-142 levels correspond to the green light that allows the initiation of differentiation (Fig 8A).

The mESC heterogeneity we describe here is caused by the action of a single microRNA. Such cell-to-cell variation is inaccessible to single-cell mRNA-Seq approaches and undetectable by traditional biochemical assays which yield population-average data. Our

findings add a previously uncharacterized layer of cellular heterogeneity that is upstream of the so far reported mESC heterogeneity at the level of transcription factors (Chambers *et al*, 2007; Singh *et al*, 2007; Toyooka *et al*, 2008).

The differential signaling pathway activation in the "high" and "low" miR-142 state explains the functional phenotypic differences we observed. High AKT activation in "low" miR-142 mESCs is likely to render them more clonogenic, as AKT signaling is known to facilitate cell survival at clonal density (Paling *et al*, 2004). The higher sensitivity of "low" miR-142 mESCs to instructive differentiation cues can be readily explained by their non-repressed activation of ERK signaling, a major transmitter of differentiation stimuli (Kunath *et al*, 2007). Indeed, the genetic deletion of *mir142* led to increased levels of ERK and AKT activation, which explains the superior performance of $mir142^{-/-}$ mESCs in differentiation and clonogenicity assays.

The action of miR-142 in controlling competence for differentiation might be of functional relevance in *in vivo* development or in the maintenance of adult stem cell compartments. Indeed, miR-142 is required to specify hemangioblast fate (Nimmo *et al*, 2013) and enforced miR-142 expression impairs macrophage differentiation *in vivo* (Sonda *et al*, 2013). Moreover, $mir142^{-/-}$ mice have numerous hematopoietic defects: impaired lymphopoiesis (Kramer *et al*, 2015), megakaryopoiesis (Chapnik *et al*, 2014) and $CD34^+$ dendritic cell homeostasis (Mildner *et al*, 2013).

Stochasticity in cell fate decisions (Losick & Desplan, 2008) has been described to play an important role ranging from bacterial differentiation (Süel *et al*, 2006) via cancer development (Gupta *et al*, 2011) to mammalian stem cells (Klein *et al*, 2010). However, the *mir142* stochastic switch we uncovered here is not a fate switch but rather a mechanism that determines the competence of stem cells to

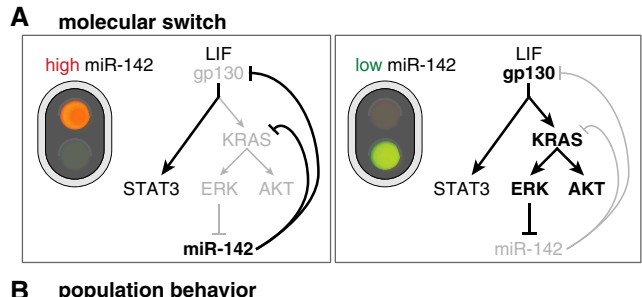

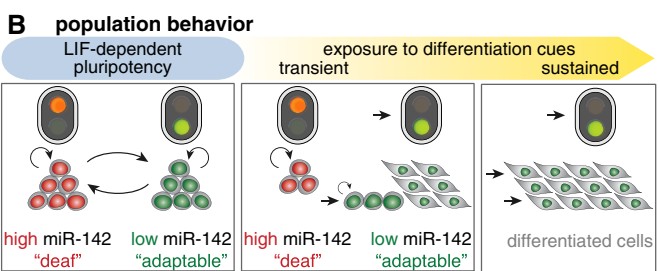

**Figure 8. The bimodal regulation of miR-142 gates the exit from pluripotency.**

A Illustration of the miR-142–Ras/ERK signaling bistable system as the molecular wiring of a traffic light.

B Stochastic switching of individual cells between a state responsive and a state deaf to signaling changes leads to phenotypic diversification at the population level.

respond to external stimuli. Cells that have entered the "high" miR-142 state are deaf to instructive differentiation signals and are therefore locked in an undifferentiated state until they interconvert into the "low" miR-142 state. We suggest that maintaining a stem cell subpopulation in a "high" miR-142 state irresponsive to differentiation cues is a mechanism to protect a reservoir of stem cells from differentiation, which can slowly interconvert into a responsive "low" miR-142 state without being completely diminished due to the high rate of self-renewal (Fig 8B). Such a buffering mechanism safeguards developmental plasticity in the face of fluctuating or noisy environmental conditions and would prevent depletion of a stem cell reservoir even upon long periods of exposure to differentiation signals.

Our results explain why clonal stem cells fail to uniformly differentiate even in a uniform differentiation culture environment (Canham *et al*, 2010). Therefore, these findings are of fundamental interest to regenerative medicine applications where induced pluripotent stem cells failing to differentiate pose a risk of tumor development (Cohen & Melton, 2011).

# Materials and Methods

### Construct design

All constructs were cloned using the MXS-chaining approach (Sladitschek & Neveu, 2015). Primer sequences are available upon request from the authors.

#### miRNA reporter constructs

PCR-amplified DNA fragments encoding the open reading frame of H2B-mCherry and H2B-Citrine were placed on either side of a bidirectional promoter. The PGK-promoter-based version consisted of four PGK enhancer elements between two back-to-back oriented minimal PGK promoters (McBurney *et al*, 1991). The version based on the CAG-promoter (Niwa *et al*, 1991) consisted of four CMV immediate early enhancer elements between two back-to-back arranged fragments containing the first exon and partial intron of chicken β-actin gene linked to the splice acceptor of the rabbit β-globin gene. The rabbit bGpA was used for both fluorescent proteins, but a binding site, perfectly complementary to the miRNA to be monitored was incorporated 11 bp downstream of the Citrine stop codon. A *PGK::hygroR-bGHpA* cassette was included for selection. For use in the *Rex1*-dGFP knockin mESC line (kindly provided by Austin Smith), we constructed an activity reporter based on the same bidirectional CAG-promoter driving the expression of H2B-2xTagBFP as normalizer and H2B-Cherry as detector. Stable mESCs lines expressing the activity reporter were generated.

#### mir142 transcriptional reporter

The 1.52 kb fragment upstream of the *mir142*-hosting lincRNA gene (ENSMUSG00000084796) was PCR-amplified from mouse genomic DNA from 129 genetic background and cloned in front of *H2B-TagBFPx3-bGpA*. The plasmid contained a *PGK::neoR-bGHpA* cassette for selection. A stable transgenic mESC line expressing the *mir142* transcriptional reporter was established in a line stably expressing the miR-142 activity reporter.

#### Inducible mir142 construct

The lincRNA containing gene (ENSMUSG00000084796) was PCR-amplified from mouse genomic DNA from 129 genetic background and cloned on one side of a bidirectional Tet-promoter. On the other side of the promoter, we cloned *NLS-TagBFPx3-PEST2D-bGHpA* to quantify the induction level. The plasmid contained a *PGK::neoR-bGHpA* selection cassette and a *PGK::rtTA-bGHpA* cassette. A stable cell line (in a background stably expressing the miR-142 activity reporter) expressing *mir142* in the presence of doxycycline at a dosage corresponding to 1.5 times the levels of endogenous *mir142* expression in "high" miR-142 state cells was used as *mir142* gain-of-function mESCs. Total miR-142 levels in those cells represented 0.05% of the total miRNA pool and were similar to the endogenous miR-142 expression levels found in the top 10% "high" miR-142 state cells, ruling out any overload of the miRNA biogenesis machinery. The *mir142* stem loop-encoding fragment was replaced by a control miRNA stem loop in the control plasmid and a stable transgenic mESC line containing the construct was established in a line stably expressing the miR-142 activity reporter.

### Cell culture

Mouse ESCs (R1 provided by the EMBL Transgenic Service or E14tga2, a kind gift of Michael Elowitz) were maintained without feeders in "LIF+serum" medium (DMEM high glucose, no glutamine, with sodium bicarbonate, Invitrogen) supplemented with 15% ES-qualified EmbryoMax Fetal Calf Serum (Millipore), 10 ng/ml murine LIF (EMBL Protein Expression and Purification Core Facility), 1× Non-Essential Amino Acids, 2 mM L-glutamine, 1 mM sodium pyruvate, 100 U/ml penicillin and 100 μg/ml streptomycin, 0.1 mM 2-mercaptoethanol (all Invitrogen) on culture dishes (Nunc) coated with 0.1% gelatin (Sigma) solution and cultured at 37°C with 5% $CO_2$. N2B27 medium was prepared from a 1:1 mixture of DMEM/F12 (without HEPES, with L-glutamine) and neurobasal medium with 0.5× B-27 (without vitamin A) and 0.5× N-2 supplements, 100 U/ml penicillin and 100 μg/ml streptomycin, 0.25 mM L-glutamine, 0.1 mM 2-mercaptoethanol (all Invitrogen), 10 μg/ml BSA fraction V and 10 μg/ml human recombinant insulin (both Sigma). "2i" medium (Ying *et al*, 2008) was N2B27 medium supplemented with 3 μM CHIR99021 and 1 μM PD0325901 (both Tocris Bioscience). "LIF+BMP4" (Ying *et al*, 2003a) medium was prepared by adding 10 ng/ml murine BMP4 (R&D Systems) to N2B27 medium with 10 ng/ml murine LIF. Cells under primed pluripotency conditions (Greber *et al*, 2010) were cultured in N2B27 medium with 12 ng/ml murine FGF2 and 20 ng/ml murine Activin A (both PeproTech). Medium was changed daily and cells were passaged every other day with 0.05% Trypsin-EDTA or StemPro Accutase (Invitrogen) at a passaging ratio of 1/3–1/12. mESCs were differentiated to endoderm precursors, mesoderm precursors and neuroectoderm precursors as described (Ying *et al*, 2003b; Borowiak *et al*, 2009; Torres *et al*, 2012).

### Generation of *mir142*[+/−], *mir142*[−/−] mESCs and rescue of *mir142* knockout

RNA-guided Cas9 nucleases were used to delete the *mir142* gene. Two guide RNA inserts (with genome target sequences:

5′-GGTGGCCTGAAGAATCCCCG, 5′-GGAGCCATGAAGGTCTTTCG) were designed and cloned in pX330-U6-Chimeric-BB-CBh-hSpCas9 following Hsu *et al* (2013). Both Cas9 plasmids were cotransfected in mESCs stably expressing the miR-142 activity reporter. Successfully edited clones corresponding to $mir142^{+/-}$ and subsequently to $mir142^{-/-}$ were identified by the derepression of the miR-142-3p activity reporter and the deletion was confirmed by sequencing of genomic PCR products. For the $mir142^{-/-}$ rescue construct, the 1.52 kb upstream region and the *mir142*-hosting lincRNA gene (ENSMUSG00000084796) were combined and a *PGK::puroR-bGHpA* cassette was added for selection. Stable transgenic mESC lines expressing the construct were generated in a $mir142^{-/-}$/miR-142 activity reporter background.

### Generation of $Kras^{-/-}$ mESCs and $Kras^{\Delta142/\Delta142}$ mESCs

RNA-guided Cas9 nucleases were used to delete the *Kras* gene. Four guide RNA inserts targeting the first exon of *Kras* which is essential (Johnson *et al*, 1997) (with genome target sequences: 5′-TATACTCAGTCATTTTCAGC, 5′-GACTGAGTATAAACTTGTGG, 5′-CTGAATTAGCTGTATCGTCA, 5′-GCTAATTCAGAATCACTTTG) were designed and cloned in pX330-U6-Chimeric-BB-CBh-hSpCas9 following Hsu *et al* (2013). The four Cas9 plasmids were cotransfected in mESCs stably expressing the miR-142 activity reporter. Successfully edited clones corresponding to $Kras^{-/-}$ were validated by checking for the deletion of the first exon by genomic PCR and the absence of any KRAS protein expression by Western blot. miR-142 bindig sites in the 3′-UTR of the *Kras* gene were deleted using RNA-guided Cas nucleases. Nine guide RNAs targeting the three miR-142 sites (with genome target sequences: 5′-TGATAAAGTCTAGGACACGC, 5′-TTATCATCTTTCAGAGGCGT, 5′-TAGACTTTATCATCTTTCAG, 5′-TTGAGATTGAATTGTTGTAG, 5′-ACAATTCAATCTCAATCCTT, 5′-GCAGCAGGAATGAAGTCCAA, 5′-GTTTGATTTAAAAGTTGCAT, 5′-CATTTCAAAAAAATAGTTTA, 5′-TAATTATTTCATTTTTCTAA) were designed and cloned in pX330-U6-Chimeric-BB-CBh-hSpCas9 following Hsu *et al* (2013). Successfully edited clones corresponding to $Kras^{\Delta142/\Delta142}$ mESCs were isolated after cotransfection of the nine Cas9 plasmids in mESCs stably expressing the miR-142 activity reporter.

### Flow cytometry and fluorescence-activated cell sorting

Cells were analyzed on an LSRFortessa flow cytometer (BD BioSciences). FACS purification was carried out using two MoFlo sorters (DakoCytomation) or a BD Influx sorter (BD BioSciences).

### Pharmacology

mESCs were cultured in LIF+serum supplemented with the following compounds at the indicated concentrations: AKT activator SC 79 (Tocris) at 5 μM, MEK inhibitor PD0325901 (Tocris) at 0.2–1.2 μM, STAT3 inhibitor SC-1 (Sigma) at 5 μM and GSK-3 inhibitor CHIR99021 (Tocris) at 3 μM. DMSO served as vehicle and as negative control in all cases.

### Immunoblot and immunostainings

Primary and secondary antibodies are described in the Appendix.

### RNA-seq library construction

RNA was extracted from cells trypsinized from plates or pelleted directly after FACS-sorting using the MirVana kit (Ambion) following the manufacturer's instructions. miRNA and mRNA libraries were prepared from the same total RNA sample. Twenty-one barcoded miRNA libraries were prepared using NEBNext Multiplex Small RNA Library Prep Set for Illumina (New England Biolabs) following the manufacturer's instructions. Seventy-eight barcoded mRNA libraries were prepared using TruSeq RNA Sample Preparation (Illumina) following the manufacturer's instructions. Libraries were run on Illumina HiSeq 2000 in the 50SE regime. Sequencing results are available on ArrayExpress with accession E-MTAB-2830, E-MTAB-3234 for mRNA-seq and E-MTAB-2831 for miRNA-seq.

### RNA-seq analysis

We built a Bowtie index for Ensembl cDNAs of the mouse genome release GRCm38 masked with RepeatMasker (Smit, AFA, Hubley, R and Green, P. RepeatMasker Open-3.0. 1996–2010 http://www.repeatmasker.org). mRNA reads were aligned to this index using Bowtie (Langmead *et al*, 2009) with default parameters. mRNA read counts were determined for each Ensembl ID using custom Python scripts. Read counts were not normalized by the transcript length for individual genes as we were solely interested in relative expression changes across samples. After trimming, miRNA reads were matched to miRBase release 19 (Griffiths-Jones *et al*, 2008) mouse sequences allowing no mismatch using custom Python scripts. Read counts were normalized to account for different sequencing depth. The normalization factor was determined by matching median-filtered log-transformed read counts for two samples to the identity line. For hierarchical clustering analysis, we kept genes with a maximal expression > 1 transcript per cell across samples and at least a four-fold variation in expression and used Pearson's correlation coefficient as a distance. Principal component analysis was carried out as described in Neveu *et al* (2010).

### Live imaging

mESCs were seeded at single-cell density (200 cells/cm$^2$) onto gelatin-coated Lab-Tek glass-bottom chamber slides (Nunc) or μ-slides (Ibidi). Confocal sections were acquired on an inverted SP8 confocal microscope (Leica) equipped with 40× PL Apo 1.1 W objective in an incubation chamber at 37°C under a humidified 5% $CO_2$ atmosphere. Citrine and mCherry were excited with 514- and 561-nm lasers, and green and red signals were acquired sequentially using HyD detectors. Images were segmented using custom Python scripts (see Appendix for details).

Phase contrast bright-field images were acquired on an inverted DM IL LED (Leica) microscope with HI PLAN I 10×/0.22 PH1 and HI PLAN I 20×/0.30 PH1 objectives.

### Modeling of the activity reporter response

We modeled the reporter ratio $r$ dependence on the miRNA concentration $M$ by $r = 1/(1 + M^n/K^n)$ where $K$ is the binding constant and $n$ is the Hill coefficient of the interaction. Using deep sequencing

results for the reporter transcript and miR-142-3p from the same sample, we found that $n = 1.07 \pm 0.04$, that is there is no cooperativity in the reporter response.

## Modeling of state switching

### Population behavior

The temporal dynamics of the interconvertible system of the "high" and "low" miR-142 states is governed by the following equations:

$$\frac{dH}{dt} = k_{high}H - k_1 H + k_{-1}L$$

$$\frac{dL}{dt} = k_{low}L + k_1 H - k_{-1}L$$

where $H$ and $L$ are the number of cells in the "high" and "low" miR-142 states, respectively, $k_{high}$ and $k_{low}$ are the division rates of the two states, $k_1$ the switching rate from "high" to "low" miR-142 state and $k_{-1}$ the switching rate from "low" to "high" miR-142 state. Given $k_{high} = k_{low} = k$ from experimental data, the evolution of the population reads:

$$H = \left( \frac{k_{-1}}{k_1 + k_{-1}}(H_0 + L_0)(1 - e^{-(k_1 + k_{-1})t}) + H_0 e^{-(k_1 + k_{-1})t} \right) e^{kt}$$

where $H_0$ and $L_0$ are the initial number of cells in "high" and "low" miR-142 state.

### Single-cell behavior

To model culture-reconstitution experiments from single founder cells, we allowed for stochastic state switching once per cell cycle with probability $k_1$ and $k_{-1}$ for 14 divisions (corresponding roughly to the 15,000-strong cell population analyzed in the experiment) using $k_1 = 1.5 \, k_{-1}$. We introduced a survival bias for "low" miR-142 cells compared to "high" miR-142 cells under single-cell plating conditions for the first two cell divisions. Two parameters were adjusted: $k_1 + k_{-1} = 0.08$ per cell division and the survival bias which was set to 8. State distribution was determined for 170 independently simulated colonies. Confidence intervals were determined by simulating 100 times 170 colonies for each miR-142 state.

## Modeling of the miR-142–ERK signaling double-negative feedback loop

The temporal dynamics of the double-negative feedback loop between miR-142 and LIF-induced ERK signaling was modeled by the following equations:

$$\frac{dM}{dt} = -k_{d1}M + k_1 \frac{1}{1 + E^{n_E}/K_E^{n_E}}$$

$$\frac{dE}{dt} = -k_{d2}E + k_2 \frac{1}{1 + M^{n_1}/K_1^{n_1}}(E^{tot} - E)$$

where $M$ is the concentration of miR-142, $E$ the fraction of active ERK, $k_{d1}$ and $k_{d2}$ the degradation rate of miR-142 and active ERK,

$E^{tot}$ the total concentration of ERK (assumed to be constant), $K_E$ the active ERK repression constant, $k_1$ the production rate of miR-142, $k_2$ the maximum activation rate of ERK, $n_E$ the Hill coefficient of ERK-mediated miR-142 repression and $n_1$ the Hill coefficient of the miR-142-mediated ERK repression. ERK is known to dimerize (Khokhlatchev *et al*, 1998) so $n_E = 2$. miR-142 represses multiple components of the LIF-induced MEK/ERK cascade so $n_1 > 1$. We took $n_1 = 2$ without any loss of generality of the findings. By introducing the parameters $\alpha = k_1/K_1 k_{d1}$, $\beta = E^{tot}/K_E$, $\gamma = k_2/k_{d2}$, and introducing the rescaled quantities $X = M/\alpha K_1$ and $Y = E/E^{tot}$, we can rewrite the system as a function of $X$ and $Y$:

$$\frac{dX}{dt} = -k_{d1}X + \frac{k_{d1}}{1 + \beta^2 Y^2}$$

$$\frac{dY}{dt} = -k_{d2}Y + \frac{k_2}{1 + \alpha^2 X^2}(1 - Y)$$

$\alpha$ represents a rescaled miR-142 turnover and $\gamma$ an ERK activation turnover. The nullclines are:

$$Y = \frac{\gamma}{1 + \gamma + \alpha^2 X^2}$$

$$Y = \frac{1}{\beta}\sqrt{\frac{1}{X} - 1}$$

$mir142^{+/-}$ cells are equivalent to having a miR-142 production rate of $k_1/2$. We determined numerically steady state solutions in the $\alpha$, $\beta$ and $\gamma$ parameter space. Theoretical phase diagram (Fig 7G) is shown for $\beta = 10$. miR-142 expression levels (Fig 7H) are computed for $0 \leq \alpha \leq 20$, $\beta = 10$, $\gamma = 1$.

Nullclines (Fig 7K) are computed for $\alpha = 12$, $\beta = 10$, $\gamma = 1$.

## Statistical analysis

Statistical tests were computed using the Python SciPy module. When appropriate, we corrected for multiple hypothesis testing following Benjamini and Hochberg (1995).

## Data deposition

Sladitschek and Neveu (2015) A toggle-switch between miR-142 and LIF-signaling creates heterogeneity among mouse ES cells. ArrayExpress E-MTAB-2830, E-MTAB-2831 and E-MTAB-3234.

Expanded View for this article is available online.

## Acknowledgements

We thank Alexis Perez Gonzalez for expert advice on flow cytometry and FACS-related experiments. We thank Jan Ellenberg and Ana Martin-Villalba for comments on the manuscript. We thank Feng Zhang for providing pX330-U6-Chimeric_BB-CBh-hSpCas9 (Addgene plasmid 42230) and Austin Smith for providing the *Rex1*-dGFP mESC line. This work was technically supported by the EMBL Genomics Core facility and Flow Cytometry Core facility. The study was funded by EMBL.

## Author contributions

HLS and PAN conceived the study. HLS carried out the experiments. HLS and PAN analyzed the data. HLS and PAN wrote the manuscript.

## Conflict of interest

The authors declare that they have no conflict of interest.

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
