## [Review Process File · Molecular Systems Biology]

The bimodally expressed microRNA miR-142 gates exit from pluripotency

Hanna L. Sladitschek and Pierre A. Neveu

Corresponding author: Pierre A. Neveu, European Molecular Biology Laboratory

Review timeline:

Submission date:	24 August 2015
Editorial Decision:	22 October 2015
Revision received:	17 November 2015
Accepted:	24 November 2015

Editor: Thomas Lemberger

Transaction Report:

1st Editorial Decision

22 October 2015

Thank you again for submitting your work to Molecular Systems Biology and apologies for the delay in getting back to you, which was due to the late arrival of the referee reports. We have now heard back from the two referees who agreed to evaluate your manuscript. As you will see from the reports below, the referees find the topic of your study of potential interest. They raise, however, several concerns and questions, which should be convincingly addressed in a revision of the present work.

Some of the key points include:

- the need of a deeper analysis of the RNAseq data.
- the need to have more direct mechanistic evidence for the double negative KRAS-miR142 feedback loop. Given that KRAS seems to be a direct target for miR142, abolition of the miR target site seems to be the most promising approach.

If you feel you can satisfactorily deal with these points and those listed by the referees, you may wish to submit a revised version of your manuscript. Please attach a covering letter giving details of the way in which you have handled each of the points raised by the referees. A revised manuscript will be once again subject to review and you probably understand that we can give you no guarantee at this stage that the eventual outcome will be favorable.

REFeree COMMENTS

Reviewer #1:

There have been considerable efforts to understand how embryonic stem cell pluripotency is regulated by transcription factors (e.g. Oct4, Sox2, Nanog), but comparatively less analysis of microRNA regulation. This study examined whether a panel of candidate miRNAs exhibited interesting expression patterns in mouse ES cells and found one, miR-142, that showed bimodal expression. Through a series of logical and straightforward studies, combined with modeling, they concluded that this miRNA serves as a negative regulator of Akt and MAPK signaling and thereby represses cell differentiation. In turn, MAPK apparently represses the expression of this microRNA, constituting a double negative feedback loop that could underlie the bimodal (i.e. potentially bistable) expression profile of miR-142. This is an interesting study that establishes the importance of a miRNA regulator of pluripotency, a novel finding. In addition, the mathematical analysis adds further to the paper. There are some questions, however.

If miR-142 plays an important role in maintaining pluripotency, one would imagine that miR-142 null mice would exhibit major defects in development. So it is somewhat surprising that such null mice live to adulthood and exhibit solely (or primarily) hematopoietic deficits. Is it known whether miR-142 is expressed/active in the inner cell mass and why its loss wouldn't lead to premature cell differentiation and developmental deficits in the early embryo?

There are many microRNAs that expressed in ES cells, as established in prior literature and in Figure 1F. Given that miR-142 apparently functions within in signaling pathways that regulate the pluripotency circuit, it is surprising that when comparing cells with high and low miR-142 levels, only expression of this miRNA and no others is different. For example, low Rex expressing cells were also low in miR-142 expression (though not all miR-142 low cells were Rex low), and one would expect variable miRNA expression levels in cells with variable Rex levels. What are potential reasons that other miRNAs aren't differentially expressed in miR-142 low/high cell populations?

In Figure 1G, can the authors clarify what the different data points are? Does each correspond to a different cell population sorted by FACS for different levels of miR reporter expression level?

The authors observed that cells interconverted between low and high miR expression levels at a slow/stochastic rate. They also observed that cells in a high miR expression state exhibited delayed differentiation. Were the delays in differentiation consistent with the rate constant for switching from high to low miR-142 level? That is, did differentiation occur on a time scale that would be consistent with high miR cells stochastically switching off at the previously measured rate, then undergoing differentiation? Or did differentiation conditions accelerate the rate at which cells switched to low miR-142 levels?

The authors found that the rates of stochastic switching could account for the distribution of low/high cells in a colony for cells that initially started in the low state but not in the high state. They hypothesized this was due to differences in cell survival for low/high miR-142 expression levels, which they showed using a clonality assay. However, single cell survival to generate a clone can easily be different from individual cell survival within a larger population, due to paracrine and cell-cell adhesion effects. Does the survival of low vs. high miR-142 cells also differ within colonies or more dense cultures? And what could mechanistically account for such a difference in survival?

The authors found that miR-142 regulated gp130 and Kras levels, which likely led to differences in Akt and MAPK activity. Why wasn't STAT3 also affected?

Finally, the model showed proof of concept that a double negative feedback loop could lead to the bistable expression profile. It is well known and even intuitive that double negative loops can generate bistability, so what's almost more important is whether it does so for reasonable values of kinetic parameters. The authors should spend more time/effort discussing where the parameter values came from, which ones are fits/fudges, and whether the fit parameters have physiologically reasonable values.

In summary, this is an interesting blend of experimentation with some modeling to discover and establish the functional importance of a miR that apparently regulates mES pluripotency in culture. The work is interesting and will be of interest to readers once a number of questions can be addressed.

Reviewer #2:

This is a very nice piece of work that should be published. This work will primarily interest stem cell biologists, but the example of a bistable system will also make the work interesting to molecular systems biologists. The novelty in stem cell biology is high; there is no really new novel concepts in molecular systems biology, but the work is done at fairly high quality.

Below I have a number of suggestions but no truly major issues.

My overall summary of the results are as follows:

The paper identifies bimodal expression of miR-142 in mouse embryonic stem cells grown in LIF+serum, and heterogeneous (but not quite bimodal) miR-142 expression in the cells grown in LIF+2i conditions. The paper argues that this bimodality reflects a double-negative feedback loop between miR-142 and ERK signaling via KRas, and that the subsequent effect on ERK and AKT activity lead to differences in response of mES cells to differentiation cues. The degree to which all of this is relevant to in vivo developmental biology is not clear, but that is the case for much mES cell biology. As a pure in vitro phenomenon this a very interesting finding, and the work is carefully executed. In more detail: the paper shows that miR-142 is uncorrelated with heterogeneity in key pluripotency factors, and that the two populations interconvert. Differentiation of the ES cells then shows that miR-142-expressing cells are delayed in differentiation compared to miR-142-low cells, and that constitutive miR-142 expression blocks differentiation. A mechanism is explored relating miR-142 activity to ERK/AKT pathway activity. The evidence for this mechanism is solid: ERK/AKT activity inversely correlates with miR-142 expression, as does gp130/Kras expression; constitutive ERK activation represses miR-142, and constitutive miR-142 expression represses ERK activity and gp130/Kras levels, while miR-142 knock-down has the opposite effect. These results show indirect evidence of a double-negative feedback between miR-142 and ERK activity. Drawing on the well-known feature of double-negative feedback loops that is the capacity of these systems to exhibit bistability, the authors propose that double-negative feedback explains the bimodal gene expression, specifically proposing that Kras mediates negative feedback between ERK activity and miR-142 expression. Kras knockout indeed leads to uniform miR-142 expression as expected.

Comments:

Introduction:

- The statement in the third paragraph that "which miRNAs control stem cell pluripotency decisions and by what mechanism they carry out this important function is largely unknown" is misleading. For example the let-7 miRNA is well known to regulate differentiation from the pluripotent state through a well-understood mechanism. Other examples also exist. Some of these studies should be cited in order to be fair to the field. Or the claim should be more specific about what we do not understand about miRNAs in pluripotency.

miR-142 is a new marker of mESC heterogeneity under naïve pluripotency conditions:

- The validation of the miR-142 reporter, and the existence of high and low activity states in mESCs, is careful and convincing.

The two miR-142 states are indistinguishable by pluripotency markers:

- The authors claim that asymmetry along the miR-142 activity axis may represent novel subpopulations within the naïve pluripotent state. This is an interesting idea, but in our opinion to support this claim it should be shown that miR-142 high and low states do not correspond to other known subpopulations in naïve mouse ES cells. For example, *Fgfr2/Fgf4* is an important axis of asymmetry in the early mouse embryo. Asymmetry along this axis precedes asymmetry in *Nanog*, *Oct4* and *Rex1*. This could be tested simply by examining the RNA sequencing data from miR-142 high and low populations, already in this paper.
- More generally, since RNA-Seq is performed, why not comment on the differences observed in RNA-Seq?
- The RNA-Seq data should be made available through GEO or another repository.

- Extending the above point, a more comprehensive comparison of other curated pluripotency markers between miR-142 high and low populations should be completed. A list of potentially interesting genes can be found in (Young et al. 2008) or more recent reviews. Optionally, the authors might also compare their RNA-Seq data to lists of heterogeneously expressed genes obtained from single cell RNA-Seq of ES cells, e.g. (Klein et al., 2015) or (Islam et al., 2013).
- The analysis would be clearer than the authors existing Figure 2b, which is currently slightly difficult to interpret since only 4 interesting genes are highlighted and the rest of the genes are so dense on the presented log scale that potentially interesting asymmetries between the high and low subpopulations in other pluripotency genes is hard to see.

The two miR-142 states interconvert stochastically:

- To fit the stochastic fate-switching model to the experimental data from clonogenic assays in Figure 4, the authors introduce a survival bias parameter for high versus low miR-142 single-cells. This model produces a better fit to the data, but other models might be imagined that would also produce a better fit just by making the model more flexible. So the precise model chosen isn't obviously the right choice, but there probably isn't room in this paper to more carefully explore what is going on during state interconversion, and in any case the general conclusions are not going to be changed by further refinement here.

Constitutive miR-142 expression locks cells in an undifferentiated state

- The blocked differentiation phenotype of miR-142 constitutive expression cells is striking; it would be helpful to provide quantitative statistics on multiple colonies, rather than just showing selected colonies. I imagine that the authors already have multiple colony images from their initial experiments so hopefully only further data analysis is required. This would give confidences that the frames presented in the immunostaining images are representative.

The "high" mir142 subpopulation is delayed in differentiation

- In this section the authors show that native variation in miR-142 within a population of naïve mESCs influences their propensity to differentiate. High miR-142 expression blocks differentiation while low expressing cells differentiate freely. The exact same comment applies as in the previous section.

miR-142 states differ in AKT and ERK activation;

- The evidence showing that miR-142 states differ in AKT and ERK activation on average is convincing. It could be an interesting exercise to extend these results by performing FACS analysis of pERK and pAKT in the miR-142 high versus low populations, to determine whether these differences are uniform across each of the two miR-142 states, or interesting variability within each state. This does not seem necessary for publication however.
- The authors explore the connection between miR-142 AND pathways downstream of LIF by using chemical inhibitors of each of the DOWNSTREAM pathways. They produce convincing evidence that ERK activity affects miR-142 expression in a dose dependent fashion. An unexplored axis in these experiments is the effect of forced activation of the ERK, AKT or STAT3 pathways, for example by overexpressing each of these three signaling molecules. This could reveal additional regulation not currently observed by the authors. But this also does not seem necessary for publication.

miR-142 balances the AKT and ERK activity

- No comments.

The miR-142-ERK double-negative feedback loop creates a bistable system

- The authors propose a double-negative feedback loop to generate a bistable system, and for the readers' convenience they recreate the well known phase diagrams for this system.
- The tests of the model show that when KRAS is eliminated, high miR-142 expression dominates, while if miR-142 is depleted bimodal expression still persists. Although consistent with the model, these tests do not establish directly that the double negative feedback loop exists.
- The authors could much more cleanly test this by eliminating binding site for miR-142 in KRAS, and eliminating binding site for ERK in the miR-142 promoter

- As a challenge, can the authors imagine other models that are either consistent with the data or ruled out by the data? The double-negative feedback is a canonical explanation for bistability, and one is left wondering if other explanations are being overlooked.

Minor comments:

- Typo on pg. 4, 3rd last line, low miR-142 "sate", should read "state". The sentence should also indicate that the high and low states are only observed together in the REX1 high compartment, as the low miR-142 state is present individually in the REX1 low compartment.
- Overall the writing is lucid and clear. And the figures are attractive and easy to interpret.

1st Revision - authors' response

17 November 2015

Reviewer #1:

There have been considerable efforts to understand how embryonic stem cell pluripotency is regulated by transcription factors (e.g. Oct4, Sox2, Nanog), but comparatively less analysis of microRNA regulation. This study examined whether a panel of candidate miRNAs exhibited interesting expression patterns in mouse ES cells and found one, miR-142, that showed bimodal expression. Through a series of logical and straightforward studies, combined with modeling, they concluded that this miRNA serves as a negative regulator of Akt and MAPK signaling and thereby represses cell differentiation. In turn, MAPK apparently represses the expression of this microRNA, constituting a double negative feedback loop that could underlie the bimodal (i.e. potentially bistable) expression profile of miR-142. This is an interesting study that establishes the importance of a miRNA regulator of pluripotency, a novel finding. In addition, the mathematical analysis adds further to the paper. There are some questions, however.

We would like to thank the reviewer for the positive evaluation and we provide a point-by-point reply to the specific comments below.

If miR-142 plays an important role in maintaining pluripotency, one would imagine that miR-142 null mice would exhibit major defects in development. So it is somewhat surprising that such null mice live to adulthood and exhibit solely (or primarily) hematopoietic deficits. Is it known whether miR-142 is expressed/active in the inner cell mass and why its loss wouldn't lead to premature cell differentiation and developmental deficits in the early embryo?

The reviewer raises an interesting question. The datasets of Tang et al., (2010) and Ohnishi et al., (2010) establish that miR-142 is expressed at low levels in the mouse preimplantation blastocyst. The mir142 knock-out has a partially penetrant phenotype during embryogenesis as one third of mir142 null pups die perinatally (Chapnik et al., 2014). Indeed, it seems surprising that there are no reported defects at early stages of embryogenesis. However, there is no strong requirement of a functional LIF pathway during normal preimplantation development: gp130^{-/-} can develop past gastrulation before dying of a variety of defects at E12-E16 (Yoshida et al., 1996). Nonetheless, gp130 is absolutely necessary for mouse embryonic stem cell (mESC) self-renewal in LIF+serum and in vivo to maintain the inner cell mass of blastocysts arrested in diapause (Yoshida et al., 1994, Nichols et al., 2001). Hence, there seems to be different requirements to maintain a pluripotent mESC population in vitro and a transient inner cell mass population in vivo.

There are many microRNAs that expressed in ES cells, as established in prior literature and in Figure 1F. Given that miR-142 apparently functions within in signaling pathways that regulate the pluripotency circuit, it is surprising that when comparing cells with high and low miR-142 levels, only expression of this miRNA and no others is different. For example, low Rex expressing cells were also low in miR-142 expression (though not all miR-142 low cells were Rex low), and one would expect variable miRNA expression levels in cells with variable Rex levels. What are potential reasons that other miRNAs aren't differentially expressed in miR-142 low/high cell populations?

The mRNA expression profile is very similar in high and low miR-142 mESCs. That might explain why the expression of most miRNAs is highly similar between the two miR-142 states. As the reviewer correctly hypothesizes, we find variable miRNA expression in cells with variable Nanog/Rex1 levels. This data is now displayed in Appendix Figure S2 and discussed page 4, paragraph 2. However in our culture conditions, only a small fraction (2.5%) of low miR-142 cells are Rex1 low while the overwhelming majority is Rex1 high. Therefore, the Rex1 low fraction contributes marginally to the average expression profile of the low miR-142 state subpopulation.

In Figure 1G, can the authors clarify what the different data points are? Does each correspond to a different cell population sorted by FACS for different levels of miR reporter expression level?

The data points in Figure 1G correspond to FACS-purified cell populations. The figure legend has been modified to mention this.

The authors observed that cells interconverted between low and high miR expression levels at a slow/stochastic rate. They also observed that cells in a high miR expression state exhibited delayed differentiation. Were the delays in differentiation consistent with the rate constant for switching from high to low miR-142 level? That is, did differentiation occur on a time scale that would be consistent with high miR cells stochastically switching off at the previously measured rate, then undergoing differentiation? Or did differentiation conditions accelerate the rate at which cells switched to low miR-142 levels?

The reviewer raises an interesting point. High miR-142 cells sustain high miR-142 expression for three days under differentiation conditions, consistent with the high miR-142 state being a stable state. After this period, high cells switch in majority to the low miR-142 state in the three differentiation regimes that we assessed. Moreover, the switching rate from low to high miR-142 state is negligible under differentiation conditions.

Therefore, after prolonged exposure to differentiation cues, switching is no longer stochastic but rather deterministic, presumably due to an absence of LIF that supports the feedback loop. In addition, after a delay of three days, switching from the high to the low miR-142 state occurs at a higher rate (~1 per day) than under pluripotency conditions.

The authors found that the rates of stochastic switching could account for the distribution of low/high cells in a colony for cells that initially started in the low state but not in the high state. They hypothesized this was due to differences in cell survival for low/high miR-142 expression levels, which they showed using a clonality assay. However, single cell survival to generate a clone can easily be different from individual cell survival within a larger population, due to paracrine and cell-cell adhesion effects. Does the survival of low vs. high miR-142 cells also differ within colonies or more dense cultures? And what could mechanistically account for such a difference in survival?

The reviewer is correct to point out that single cell survival might be different from survival under more dense conditions. Indeed, we find no difference between the survival of low and high miR-142 cells under normal passaging conditions (>10,000 cells/cm²). The difference in active AKT levels between the two states is a possible mechanistic explanation. This is in line with Paling et al. (2004) who reported that PI3K/AKT inhibition has no effect on cell survival and proliferation under regular culture conditions but affects self-renewal in clonogenicity assays.

The authors found that miR-142 regulated gp130 and Kras levels, which likely led to differences in Akt and MAPK activity. Why wasn't STAT3 also affected?

STAT3 signaling has a negative feedback implemented through the expression of its target genes of the SOCS family (notably SOCS3 in mESCs) that act as negative regulators. This could ensure a stable level of STAT3 activation. Interestingly, a specific level of STAT3 signaling –neither too low nor too high– is needed for mESC self-renewal (Niwa et al., 1998; Tai et al., 2014).

Finally, the model showed proof of concept that a double negative feedback loop could lead to the bistable expression profile. It is well known and even intuitive that double negative loops can generate bistability, so what's almost more important is whether it does so for reasonable values of kinetic parameters. The authors should spend more time/effort discussing where the parameter values came from, which ones are fits/fudges, and whether the fit parameters have physiologically reasonable values.

The referee raises a valid point. Reducing the number of parameters leaves us with only three parameters in our model. Those parameters are the following: α which is the ratio of miR-142 production and degradation rates divided by the affinity of its repressive interaction; β which corresponds to the ratio of total ERK to the affinity of its repressive interaction and γ which represents the phosphorylation/ dephosphorylation balance of ERK.

It is possible to find two steady states for $\beta > 2$ with the size of the bistability region in the (α, γ) parameter space increasing with increasing β . The aforementioned condition corresponds to total ERK levels being in excess compared to the levels necessary to exert its repressive interaction. This condition makes sense biologically.

Bistability occurs mostly for large values for α , corresponding to a small miR-142 degradation rate compared to its production rate. This is in accordance with what is known about miRNA stability. Indeed, miRNAs are stable in a wide variety of cells including mESCs (Krol et al., 2010), miRNA decay being mostly contributed by cell division (Gantier et al., 2011).

Similarly, bistability occurs mostly for large values for γ (of the order of one or greater). This corresponds to phosphorylation rates that are large compared to dephosphorylation rates, in accordance with experimental evidence (Fujioka et al., 2006).

In conclusion, only few parameters are required in our model and their values necessary to observe bistability are physiologically reasonable.

We now discuss in the manuscript those points, page 7, paragraph 3. We have added in the Appendix the Figures S11 and S13A to illustrate how the bistability region depends on β .

In summary, this is an interesting blend of experimentation with some modeling to discover and establish the functional importance of a miR that apparently regulates mES pluripotency in culture. The work is interesting and will be of interest to readers once a number of questions can be addressed.

We would like to thank the reviewer for the positive evaluation and her/his insightful comments.

Reviewer #2:

This is a very nice piece of work that should be published. This work will primarily interest stem cell biologists, but the example of a bistable system will also make the work interesting to molecular systems biologists. The novelty in stem cell biology is high; there is no really new novel concepts in molecular systems biology, but the work is done at fairly high quality.

Below I have a number of suggestions but no truly major issues.

We would like to thank the reviewer for the positive evaluation and we provide a point-by-point reply to the specific comments below.

My overall summary of the results are as follows:

The paper identifies bimodal expression of miR-142 in mouse embryonic stem cells grown in LIF+serum, and heterogeneous (but not quite bimodal) miR-142 expression in the cells grown in LIF+2i conditions. The paper argues that this bimodality reflects a double-negative feedback loop between miR-142 and ERK signaling via KRas, and that the subsequent effect on ERK and AKT activity lead to differences in response of mES cells to differentiation cues. The degree to which all of this is relevant to in vivo developmental biology is not clear, but that is the case for much mES cell biology. As a pure in vitro phenomenon this a very interesting finding, and the work is carefully

executed. In more detail: the paper shows that miR-142 is uncorrelated with heterogeneity in key pluripotency factors, and that the two populations interconvert. Differentiation of the ES cells then shows that miR-142-expressing cells are delayed in differentiation compared to miR-142-low cells, and that constitutive miR-142 expression blocks differentiation. A mechanism is explored relating miR-142 activity to ERK/AKT pathway activity. The evidence for this mechanism is solid: ERK/AKT activity inversely correlates with miR-142 expression, as does gp130/Kras expression; constitutive ERK activation represses miR-142, and constitutive miR-142 expression represses ERK activity and gp130/Kras levels, while miR-142 knock-down has the opposite effect. These results show indirect evidence of a double-negative feedback between miR-142 and ERK activity. Drawing on the well-known feature of double-negative feedback loops that is the capacity of these systems to exhibit bistability, the authors propose that double-negative feedback explains the bimodal gene expression, specifically proposing that Kras mediates negative feedback between ERK activity and miR-142 expression. Kras knockout indeed leads to uniform miR-142 expression as expected.

Comments:

Introduction:

- The statement in the third paragraph that "which miRNAs control stem cell pluripotency decisions and by what mechanism they carry out this important function is largely unknown" is misleading. For example the let-7 miRNA is well known to regulate differentiation from the pluripotent state through a well-understood mechanism. Other examples also exist. Some of these studies should be cited in order to be fair to the field. Or the claim should be more specific about what we do not understand about miRNAs in pluripotency.

The reviewer is right. We have rephrased the sentence to more specifically motivate where the novelty of our study lies. The sentence reads now (page 3, paragraph 3): "Whereas the role of transcription factor heterogeneity in defining different pluripotent substates is well established (Chambers et al, 2007; Singh et al, 2007; Toyooka et al, 2008), it is largely unknown whether such dynamic heterogeneity exists at the level of miRNA expression."

miR-142 is a new marker of mESC heterogeneity under naïve pluripotency conditions:

- The validation of the miR-142 reporter, and the existence of high and low activity states in mESCs, is careful and convincing.

The two miR-142 states are indistinguishable by pluripotency markers:

- The authors claim that asymmetry along the miR-142 activity axis may represent novel subpopulations within the naïve pluripotent state. This is an interesting idea, but in our opinion to support this claim it should be shown that miR-142 high and low states do not correspond to other known subpopulations in naïve mouse ES cells. For example, Fgfr2/Fgf4 is an important axis of asymmetry in the early mouse embryo. Asymmetry along this axis precedes asymmetry in Nanog, Oct4 and Rex1. This could be tested simply by examining the RNA sequencing data from miR-142 high and low populations, already in this paper.
- More generally, since RNA-Seq is performed, why not comment on the differences observed in RNA-Seq?

In order to get an unbiased view of gene expression differences between the two miR-142 states, we performed Gene Set Enrichment Analysis. We found no significant dysregulation of curated gene sets (canonical pathways, BioCarta and KEGG gene sets). However, we did find a difference in mRNA expression of the predicted targets of miR-142-3p: they had significantly lower expression in "high" miR-142 mESCs compared to "low" miR-142 mESCs. This new analysis is presented in Appendix Figure S4 and discussed page 4, paragraph 2. In addition, there are only marginal differences in the expression of Fgf4 and Fgfr2 between the two miR-142 states as shown in Figure EV3E.

- The RNA-Seq data should be made available through GEO or another repository.

The RNA-Seq data has already been deposited to ArrayExpress (with accession numbers E-MTAB-2830, E-MTAB-2831 and E-MTAB-3234) and will be made publicly available upon publication.

- Extending the above point, a more comprehensive comparison of other curated pluripotency markers between miR-142 high and low populations should be completed. A list of potentially interesting genes can be found in (Young et al. 2008) or more recent reviews. Optionally, the authors might also compare their RNA-Seq data to lists of heterogeneously expressed genes obtained from single cell RNA-Seq of ES cells, e.g. (Klein et al., 2015) or (Islam et al., 2013).
- The analysis would be clearer than the authors existing Figure 2b, which is currently slightly difficult to interpret since only 4 interesting genes are highlighted and the rest of the genes are so dense on the presented log scale that potentially interesting asymmetries between the high and low subpopulations in other pluripotency genes is hard to see.

We looked specifically at pluripotency markers listed in the review by Ng & Surani (2011). We found non-significant minor differences in expression of some of those genes. Their amplitude is much smaller in comparison to the differences in expression between high Nanog and low Nanog cells. This new analysis is displayed in Figure EV3E and discussed page 4, paragraph 2.

In addition, we took the list of highly variable genes from Klein et al. and compared their expression in high miR-142 cells and low miR-142 cells. We found only 14 genes out of 1891 with more than 2-fold expression changes between the two states. This number is similar to the one obtained when comparing biological replicates. In comparison, 302 genes display more than 2-fold changes in expression in low Nanog cells compared to high Nanog cells. This new analysis is displayed in Appendix Figure S3 and discussed page 4, paragraph 2.

Taken together, these new analysis support our original claims, i.e., that miR-142 expression distinguishes previously uncharacterized substates in mESCs.

The two miR-142 states interconvert stochastically:

- To fit the stochastic fate-switching model to the experimental data from clonogenic assays in Figure 4, the authors introduce a survival bias parameter for high versus low miR-142 single-cells. This model produces a better fit to the data, but other models might be imagined that would also produce a better fit just by making the model more flexible. So the precise model chosen isn't obviously the right choice, but there probably isn't room in this paper to more carefully explore what is going on during state interconversion, and in any case the general conclusions are not going to be changed by further refinement here.

We agree that there are other models that could be used to produce a better fit than our simple model without survival bias. For example, a model with equal survival probability but a switching probability that is increased at low cell density would yield a good fit of the experimental data. However, this particular model is ruled out by the finding that high miR-142 cells are less clonogenic than low miR-142 cells. Instead, the model assuming different survival bias we propose is consistent with all the experimental data. More complicated models incorporating such a survival bias could apply as well but will likely overfit the data.

Constitutive miR-142 expression locks cells in an undifferentiated state

- The blocked differentiation phenotype of miR-142 constitutive expression cells is striking; it would be helpful to provide quantitative statistics on multiple colonies, rather than just showing selected colonies. I imagine that the authors already have multiple colony images from their initial experiments so hopefully only further data analysis is required. This would give confidences that the frames presented in the immunostaining images are representative.

The reviewer raises a valid concern. In order to get quantitative unbiased data at the population level, we differentiated cells with constitutive miR-142 expression as well as mir142^{-/-} cells towards fates representing the three germ layers and assessed the distribution of Oct4 levels by quantitative immunostaining in the entire population by flow cytometry. This showed that > 95% cells with constitutive miR-142 expression were Oct4 positive whereas > 90% of mir142^{-/-} cells were Oct4 negative after 6 days of differentiation. These results are in accordance with the original images shown in Figure 5 and are displayed in Appendix Figure S6. Moreover, the deep sequencing data

shown in Figure 5G-I present an unbiased genome-wide characterization of the phenomenon using the example of endoderm differentiation. The deep sequencing data corroborate the quantitative immunostaining results.

The "high" mir142 subpopulation is delayed in differentiation

- In this section the authors show that native variation in miR-142 within a population of naïve mESCs influences their propensity to differentiate. High miR-142 expression blocks differentiation while low expressing cells differentiate freely. The exact same comment applies as in the previous section.

The reviewer raises a valid concern. In order to get quantitative unbiased data at the population level, we differentiated wild type cells towards fates representing the three germ layers and assessed the distribution of Oct4 levels by quantitative immunostaining in the entire population by flow cytometry. Cells were classified as high or low miR-142 according to their reporter ratio. This showed that > 80% high miR-142 cells were Oct4 positive whereas > 85% low miR-142 cells were Oct4 negative after 6 days of differentiation. These results are in accordance with the original images shown in Figure 6 and are displayed in Appendix Figure S8. Moreover, the deep sequencing data shown in Figure 6G-I present an unbiased genome-wide characterization of the phenomenon using the example of endoderm differentiation. The deep sequencing data corroborate the quantitative immunostaining results.

miR-142 states differ in AKT and ERK activation;

- The evidence showing that miR-142 states differ in AKT and ERK activation on average is convincing. It could be an interesting exercise to extend these results by performing FACS analysis of pERK and pAKT in the miR-142 high versus low populations, to determine whether these differences are uniform across each of the two miR-142 states, or interesting variability within each state. This does not seem necessary for publication however.

We agree with the reviewer that it would be an interesting issue to address in a follow-up study.

- The authors explore the connection between miR-142 AND pathways downstream of LIF by using chemical inhibitors of each of the DOWNSTREAM pathways. They produce convincing evidence that ERK activity affects miR-142 expression in a dose dependent fashion. An unexplored axis in these experiments is the effect of forced activation of the ERK, AKT or STAT3 pathways, for example by overexpressing each of these three signaling molecules. This could reveal additional regulation not currently observed by the authors. But this also does not seem necessary for publication.

We agree with the reviewer that it would be an interesting issue to address in a follow-up study.

miR-142 balances the AKT and ERK activity

- No comments.

The miR-142-ERK double-negative feedback loop creates a bistable system

- The authors propose a double-negative feedback loop to generate a bistable system, and for the readers' convenience they recreate the well known phase diagrams for this system.
- The tests of the model show that when KRAS is eliminated, high miR-142 expression dominates, while if miR-142 is depleted bimodal expression still persists. Although consistent with the model, these tests do not establish directly that the double negative feedback loop exists.
- The authors could much more cleanly test this by eliminating binding site for miR-142 in KRAS, and eliminating binding site for ERK in the miR-142 promoter

We deleted the miR-142 binding sites in Kras 3'-UTR using CRISPR/Cas9 and assessed the distribution of miR-142 reporter ratio in a self-renewing mESC population. As predicted by the model, the bimodal regulation of miR-142 was abrogated and all cells resided in a "low" miR-142

state. This is consistent with the absence of negative feedback on Kras. The new results are presented in Figure 7I-J and discussed page 7, paragraph 4. ERK does not directly bind to DNA. Therefore, its repressive action on mir142 expression must be mediated by either the activation of a transcriptional repressor or the degradation/inactivation of a transcriptional activator.

- As a challenge, can the authors imagine other models that are either consistent with the data or ruled out by the data? The double-negative feedback is a canonical explanation for bistability, and one is left wondering if other explanations are being overlooked.

Models relying on positive feedback loops only can generate bistability but are ruled out by the repressive action of miR-142. Our data does not exclude that some kind of cellular noise drives switching events from high to low miR-142 or low to high miR-142 states. The collapse of the bistable system to a low miR-142 state in mESCs lacking miR-142 binding sites in Kras 3'-UTR supports a direct negative feedback between miR-142 and Kras that is crucial to establish the bimodality. In addition, our data shows that ERK signaling represses mir142 expression. Thus any model of the system should include at least two negative interactions. Of course, parallel circuits and additional interactions might exist. Our goal was to incorporate only the essential interactions in the model we propose.

Minor comments:

- Typo on pg. 4, 3rd last line, low miR-142 "sate", should read "state". The sentence should also indicate that the high and low states are only observed together in the REX1 high compartment, as the low miR-142 state is present individually in the REX1 low compartment.

We have corrected the typo. Taking the reviewer's comment into account, the sentence reads now (page 4, paragraph 3): "Thus, the ``high" miR-142 state and the ``low" miR-142 state are only found together in the high Rex1 mESC compartment."

- Overall the writing is lucid and clear. And the figures are attractive and easy to interpret.